# IRF4 haploinsufficiency in a family with Whipple's disease

Antoine Guérin[1,2], Gaspard Kerner[1,2†], Nico Marr[3†], Janet G Markle[4‡],
Florence Fenollar[5‡], Natalie Wong[6,7‡], Sabri Boughorbel[3‡], Danielle T Avery[6,7‡],
Cindy S Ma[6,7‡], Salim Bougarn[3‡], Matthieu Bouaziz[1,2], Vivien Béziat[1,2],
Erika Della Mina[1,2], Carmen Oleaga-Quintas[1,2], Tomi Lazarov[8,9], Lisa Worley[6,7],
Tina Nguyen[6,7], Etienne Patin[10,11,12], Caroline Deswarte[1,2],
Rubén Martinez-Barricarte[4], Soraya Boucherit[1,2], Xavier Ayral[13], Sophie Edouard[5],
Stéphanie Boisson-Dupuis[1,2,4], Vimel Rattina[1,2], Benedetta Bigio[4], Guillaume Vogt[2],
Frédéric Geissmann[8,9,14§], Lluis Quintana-Murci[10,11,12§], Damien Chaussabel[3§],
Stuart G Tangye[6,7§], Didier Raoult[5#], Laurent Abel[1,2,4#], Jacinta Bustamante[1,2,4,15#],
Jean-Laurent Casanova[1,2,4,16,17*#]

[1]Laboratory of Human Genetics of Infectious Diseases, Necker Branch, INSERM U1163, Paris, France; [2]Imagine Institute, Paris Descartes University, Paris, France; [3]Sidra Medicine, Doha, Qatar; [4]St. Giles Laboratory of Human Genetics of Infectious Diseases, Rockefeller Branch, The Rockefeller University, New York, United States; [5]Research Unit of Infectious and Tropical Emerging Diseases, University Aix-Marseille, URMITE, UM63, CNRS 7278, IRD 198, Marseille, France; [6]Immunology Division, Garvan Institute of Medical Research, Darlinghurst, Australia; [7]St Vincent's Clinical School, Faculty of Medicine, University of New South Wales, Sydney, Australia; [8]Immunology Program, Memorial Sloan Kettering Cancer Center, New York, United States; [9]Ludwig Center, Memorial Sloan Kettering Cancer Center, New York, United States; [10]Human Evolutionary Genetics Unit, Department of Genomes and Genetics, Institut Pasteur, Paris, France; [11]CNRS UMR2000, Paris, France; [12]Center of Bioinformatics, Biostatistics and Integrative Biology, Institut Pasteur, Paris, France; [13]Rheumatology Unit, Cochin Hospital, Paris, France; [14]Weill Cornell Graduate School of Medical Sciences, New York, United States; [15]Center for the Study of Primary Immunodeficiencies, Assistance Publique-Hôpitaux de Paris, Necker Hospital for Sick Children, Paris, France; [16]Pediatric Hematology and Immunology Unit, Assistance Publique-Hôpitaux de Paris, Necker Hospital for Sick Children, Paris, France; [17]Howard Hughes Medical Institute, New York, United States

**\*For correspondence:**
casanova@rockefeller.edu (J-LC)

†These authors contributed equally to this work

‡These authors also contributed equally to this work

§These authors also contributed equally to this work

#These authors also contributed equally to this work

**Competing interest:** The authors declare that no competing interests exist.

**Abstract** Most humans are exposed to *Tropheryma whipplei* (Tw). Whipple's disease (WD) strikes only a small minority of individuals infected with Tw (<0.01%), whereas asymptomatic chronic carriage is more common (<25%). We studied a multiplex kindred, containing four WD patients and five healthy Tw chronic carriers. We hypothesized that WD displays autosomal dominant (AD) inheritance, with age-dependent incomplete penetrance. We identified a single very rare non-synonymous mutation in the four patients: the private R98W variant of IRF4, a transcription factor involved in immunity. The five Tw carriers were younger, and also heterozygous for R98W. We found that R98W was loss-of-function, modified the transcriptome of heterozygous leukocytes following

Tw stimulation, and was not dominant-negative. We also found that only six of the other 153 known non-synonymous IRF4 variants were loss-of-function. Finally, we found that *IRF4* had evolved under purifying selection. AD IRF4 deficiency can underlie WD by haploinsufficiency, with age-dependent incomplete penetrance.

DOI: https://doi.org/10.7554/eLife.32340.001

## Introduction

Whipple's disease (WD) was first described as an intestinal inflammatory disease by George H. Whipple in 1907 (*Whipple, 1907*). Its infectious origin was suspected in 1961 (*Yardley and Hendrix, 1961*), and the causal microbe, *Tropheryma whipplei* (Tw), a Gram-positive actinomycete, was detected by PCR in 1992 (*Relman et al., 1992*), and cultured in 2000 (*Raoult et al., 2000*). Tw is probably transmitted between humans via the oro-oral or feco-oral routes. WD is a chronic condition with a late onset (mean age at onset: 55 years) (*Braubach et al., 2017*) affecting multiple organs. The clinical manifestations of classical WD are arthralgia, diarrhea, abdominal pain, and weight loss (*Dobbins, 1987*; *Durand et al., 1997*; *Fleming et al., 1988*; *Mahnel et al., 2005*; *Maizel et al., 1970*). However, about 25% of WD patients display no gastrointestinal or osteoarticular symptoms, instead presenting with cardiac and/or neurological manifestations (*Durand et al., 1997*; *Fenollar et al., 2014*, *Fenollar et al., 2001*; *Gubler et al., 1999*; *Schneider et al., 2008*). WD is fatal if left untreated, and relapses occur in 2% to 33% of treated cases, even after prolonged appropriate antibiotic treatment (*Lagier et al., 2011*; *Marumganti and Murphy, 2008*). WD is rare and has been estimated to affect about one in a million individuals (*Dobbins, 1981*, *1987*; *Fenollar et al., 2007*). However, about two thousand cases have been reported in at least nine countries worldwide, mostly in North America and Western Europe (*Bakkali et al., 2008*; *Desnues et al., 2010*; *Fenollar et al., 2008a*; *Lagier et al., 2010*; *Puéchal, 2016*; *Schneider et al., 2008*). Chronic asymptomatic carriage of Tw is common in the general population, and this bacterium has been detected in feces, saliva, and intestinal mucosae. The prevalence of Tw carriage in the feces has been estimated at 2% to 11% for the general population, but can reach 26% in sewer workers and 37% in relatives of patients and carriers (*Amsler et al., 2003*; *Ehrbar et al., 1999*; *Fenollar et al., 2014*, *Fenollar et al., 2007*; *Maibach et al., 2002*; *Rolain et al., 2007*; *Schneider et al., 2008*; *Street et al., 1999*).

Seroprevalence for specific antibodies against Tw in the general population varies from 50% in France to 70% in Senegal (*Fenollar et al., 2009*, *Fenollar et al., 2014*; *Raoult et al., 2000*; *Schneider et al., 2008*). At least 75% of infected individuals clear Tw primary infections, but a minority (<25%) become asymptomatic carriers, a very small proportion of whom develop WD (<0.01%) (*Fenollar et al., 2008b*). Tw infection is therefore, necessary but not sufficient, for WD development, and it is unclear whether prolonged asymptomatic carriage necessarily precedes WD. The hypothesis that WD results from the emergence of a more pathogenic clonal strain of Tw was not supported by bacterial genotyping (*Li et al., 2008*). WD mostly affects individuals of European origin, but does not seem to be favored by specific environments. WD is typically sporadic, but six multiplex kindreds have been reported, with cases often diagnosed years apart, suggesting a possible genetic component (*Durand et al., 1997*; *Fenollar et al., 2007*; *Ponz de Leon et al., 2006*). WD patients are not prone to other severe infections (*Marth et al., 2016*). Moreover, WD has never been reported in patients with conventional primary immunodeficiencies (PIDs) (*Picard et al., 2018*). This situation is reminiscent of other sporadic severe infections, such as herpes simplex virus-1 encephalitis, severe influenza, recurrent rhinovirus infection, severe varicella zoster disease, trypanosomiasis, invasive staphylococcal disease, and viral infections of the brainstem, which are caused by single-gene inborn errors of immunity in some patients (*Andersen et al., 2015*; *Casanova, 2015a*, *Casanova, 2015b*; *Ciancanelli et al., 2015*; *Israel et al., 2017*; *Lamborn et al., 2017*; *Zhang et al., 2015*; *Ogunjimi et al., 2017*; *Vanhollebeke et al., 2006*; *Zhang et al., 2018*). We therefore hypothesized that WD might be due to monogenic inborn errors of immunity to Tw, with age-dependent incomplete penetrance.

**eLife digest**   In 1907, George Hoyt Whipple – an American pathologist working at Johns Hopkins University in Baltimore – described a new inflammatory disease that affects the intestine. Patients with this condition, now known as Whipple's disease, experience diarrhea, weight loss, and abdominal and joint pain. The disease is rare; it affects about one in a million people, mostly those over the age of 50 who are of European descent.

Later it was discovered that bacteria called *Tropheryma whipplei* cause Whipple's disease and that antibiotics can cure it. These bacteria are widespread and yet only a small minority of individuals infected with *T. whipplei* goes on to develop Whipple's disease. In some populations, over 50% of individuals have been infected with the bacteria at some point in their lives, but most will get rid of the infection. This raised the question: when exposed to the same microbe, why do some individuals develop a severe disease, while others remain unharmed?

From the 1950s onwards, scientists identified a few families with multiple members who have developed Whipple's disease. These observations suggested that human genes may play a role in determining whether a person infected with *T. whipplei* becomes ill. Rare genetic mutations that affect the immune system have also been linked to the development of life-threatening cases of influenza or tuberculosis.

Now, Guérin et al. report that, in one French family, an extremely rare mutation in the gene that codes for a protein called IRF4 may contribute to the development of Whipple's disease. This family had four members with Whipple's disease, all of whom had one copy of the gene with this rare mutation and one normal copy of the gene. The IRF4 protein acts like a switch that turns on and off some genes involved in the body's response to infection. In patients with this mutation, the IRF4 protein does not work as it should.

Guérin et al. suggest that Whipple's disease may be caused by specific genetic mutations affecting the immune system in subjects infected by *T. whipplei*. More studies are needed to see if other genetic mutations also contribute to other cases of Whipple's disease. Such studies may help physicians to better understand why some people become sick after *T. whipplei* infections while others do not. They may also help physicians to diagnose the disease, and even lead to better treatments.

DOI: https://doi.org/10.7554/eLife.32340.037

## Results

### A multiplex kindred with WD

We investigated four related patients diagnosed with WD (P1, P2, P3, and P4) with a mean age at diagnosis of 58 years. They belong to a large non-consanguineous French kindred (*Figure 1A*). The proband (P1), a 69-year-old woman, presented with right knee arthritis in 2011, after recurrent episodes of arthritis of the right knee since 1980. Tw was detected in the synovial fluid by PCR and culture, but not in saliva, feces, or small intestine tissue. Treatment with doxycycline and hydroxychloroquine was effective. At last follow-up, in 2016, P1 was well and Tw PCR on saliva and feces was negative. P2, a second cousin of P1, is a 76-year-old woman with classical WD diagnosed at 37 years of age in 1978 by periodic acid–Schiff (PAS) staining of a small intestine biopsy specimen. She was treated with sulfamethoxazole/trimethoprim. At last follow-up, in 2016, Tw PCR on saliva and feces was positive. P3, the father of P1, is a 92-year-old man with classical WD diagnosed at 62 years of age, in 1987, based on positive PAS staining of a small intestine biopsy specimen. Long-term sulfamethoxazole/trimethoprim treatment led to complete clinical and bacteriological remission. P4, the brother of P2, is a 70-year-old man who consulted in 2015 for arthralgia of the knees and right ulna-carpal joints. PCR and culture did not detect Tw in saliva and feces, but serological tests for Tw were positive. Treatment with methotrexate and steroids was initiated before antibiotics, the effect of which is currently being evaluated. All four patients are otherwise healthy. Saliva and/or feces samples from 18 other members of the family were tested for Tw (*Figure 1A*; *Figure 1—source data 1*). Five individuals are chronic carriers (mean age: 55 years) and 13 tested negative (mean age: 38 years). Nine additional relatives could not be tested. The distribution of WD in this kindred was suggestive of an AD trait with incomplete clinical penetrance.

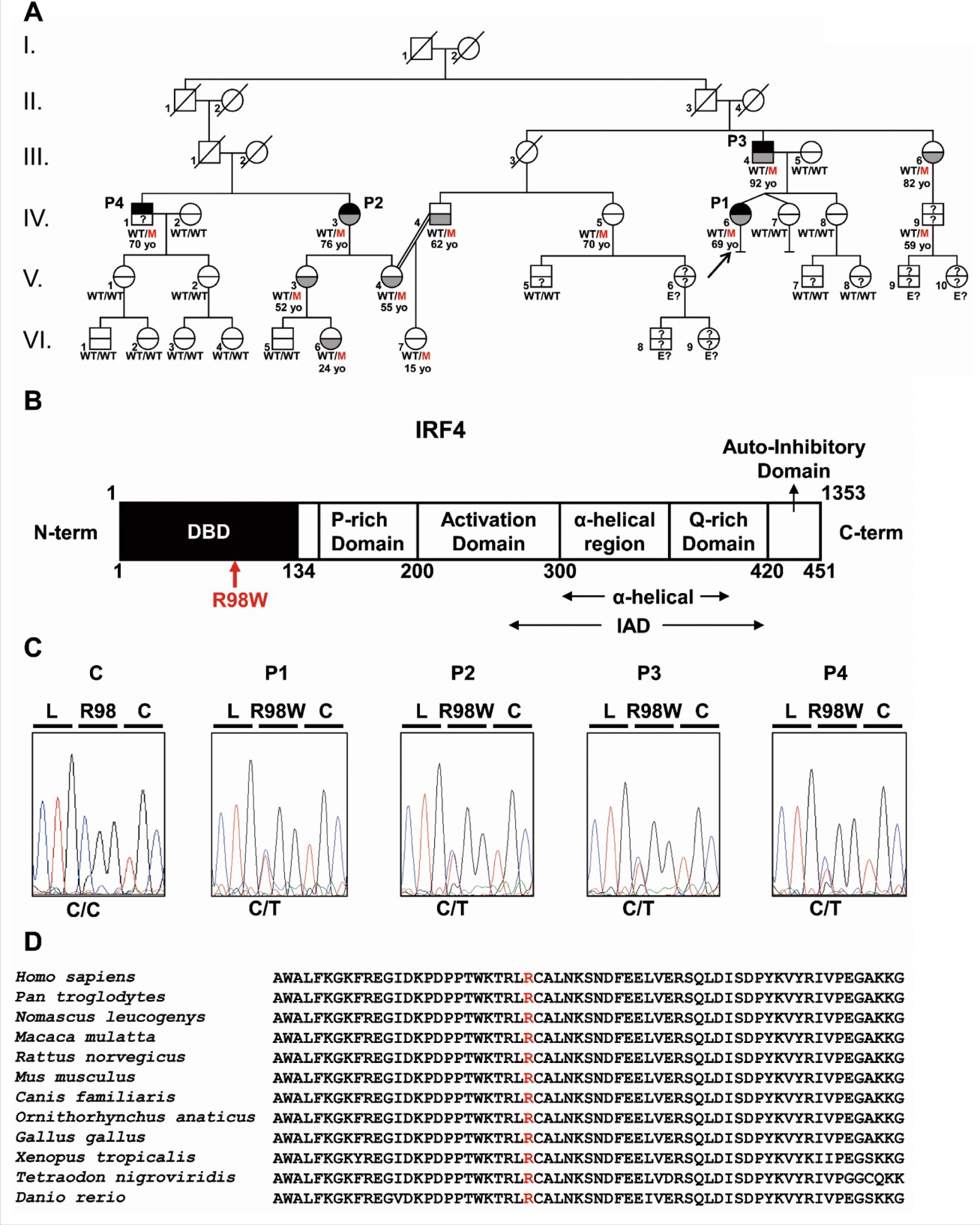

**Figure 1.** Autosomal dominant IRF4 deficiency. (**A**) Pedigree of the kindred, with allele segregation. Generations are designated by Roman numerals (I, II, III, IV, V and VI), and each individual is designated by an Arabic numeral (from left to right). Each symbol is divided into two parts: the upper part

*Figure 1 continued*

indicates clinical status for WD (black: affected, white: healthy, '?': not known); the lower part indicates whether Tw was identified by PCR (in saliva, blood, feces or joint fluid) or by PAS staining on bowel biopsy specimens (gray: Tw-positive, white: Tw-negative, '?': not tested). Whipple's disease patients are indicated as P1, P2, P3, and P4; the proband is indicated with an arrow. Genotype status and age (for *IRF4*-heterozygous individuals) are reported below the symbols. Individuals whose genetic status could not be evaluated are indicated by the symbol 'E?'. (**B**) Schematic representation of the IRF4 protein, showing the DNA-binding domain (DBD), P-rich domain, activation domain, α-helical domain, Q-rich domain, IRF association domain (IAD), and auto-inhibitory domain. The R98W substitution is indicated in red. (**C**) Electropherogram of *IRF4* genomic DNA sequences from a healthy unrelated control (C) and the patients (P1, P2, P3, P4). The R98W IRF4 mutation leads to the replacement of an arginine with a tryptophan residue in position 98 (exon 3, c.292 C > T). The corresponding amino acids are represented above each electropherogram. (**D**) Alignment of the R98W amino acid in the DBD domain of IRF4 in humans and 11 other animal species. R98 is indicated in red.

DOI: https://doi.org/10.7554/eLife.32340.002

The following source data and figure supplements are available for figure 1:

**Source data 1.** Kindred information summary.
DOI: https://doi.org/10.7554/eLife.32340.004

**Source data 2.** Non-synonymous variants within the linkage regions found in WES data from patients.
DOI: https://doi.org/10.7554/eLife.32340.005

**Figure supplement 1.** Genome-wide linkage and whole-exome sequencing analyses.

DOI: https://doi.org/10.7554/eLife.32340.003

## A private heterozygous missense IRF4 variant segregates with WD

We analyzed the familial segregation of WD by genome-wide linkage (GWL), using information from both genome-wide single-nucleotide polymorphism (SNP) microarrays and whole-exome sequencing (WES) (*Belkadi et al., 2016*). Multipoint linkage analysis was performed under an AD model, with a very rare disease-causing allele ($<10^{-5}$) and age-dependent incomplete penetrance. Twelve chromosomal regions linked to WD were identified on chromosomes 1 (x3), 2, 3, 6, 7, 8, 10, 11, 12 and 17, with a LOD score close (>1.90) to the maximum expected value (1.95) (*Figure 1—figure supplement 1A*). These regions covered 27.18 Mb and included 263 protein-coding genes. WES data analysis for these 263 genes identified 54 heterozygous non-synonymous coding variants common to all four WD patients (*Figure 1—source data 2*). Only one, a variant of the interferon regulatory factor (*IRF*) four gene encoding a transcription factor from the IRF family (*Ikushima et al., 2013*), located in a 200 kb linked region on chromosome 6 (*Figure 1—figure supplement 1A and B*), was very rare, and was even found to be private [not found in the gnomAD database, http://gnomad.broadinstitute.org, or in our in-house WES database (HGID)], whereas all other variants had a frequency >0.001, which is inconsistent with the frequency of WD and our hypothesis of a very rare ($<10^{-5}$) deleterious heterozygous allele. The variant is a c.292 C > T substitution in exon 3 of *IRF4*, replacing the arginine residue in position 98 with a tryptophan residue (R98W) (*Figure 1A, B and C*). IRF4 is a transcription factor with an important pleiotropic role in innate and adaptive immunity, at least in a few strains of inbred mice (*Shaffer et al., 2009*). Mice heterozygous for a null *Irf4* mutation have not been studied, but homozygous null mice have various T- and B-cell abnormalities and are susceptible to both *Leishmania* and lymphocytic choriomeningitis virus (*Klein et al., 2006*; *Lohoff et al., 2002*; *Mittrücker et al., 1997*; *Suzuki et al., 2004*; *Tamura et al., 2005*; *Tominaga et al., 2003*). We confirmed the *IRF4* R98W mutation by Sanger sequencing genomic DNA from the blood of the four WD patients (*Figure 1C*). Thirteen relatives of the WD patients were WT/WT at the *IRF4* locus, and 10 of these relatives (77%) tested negative for Tw carriage. Eight other relatives were heterozygous for the *IRF4* R98W mutation, five of whom (62.5%) were Tw carriers (mean age: 55 years) (*Figure 1A*; *Figure 1—source data 1*). Overall, 12 individuals from the kindred, including the four patients, the five chronic carriers of Tw, two non-carriers of Tw and one relative not tested for Tw, were heterozygous for *IRF4* R98W (*Figure 1A*; *Figure 1—source data 1*). The familial segregation of the *IRF4* R98W allele was therefore consistent with an AD pattern of WD inheritance with incomplete clinical penetrance. Chronic Tw carriage also followed an AD mode of inheritance.

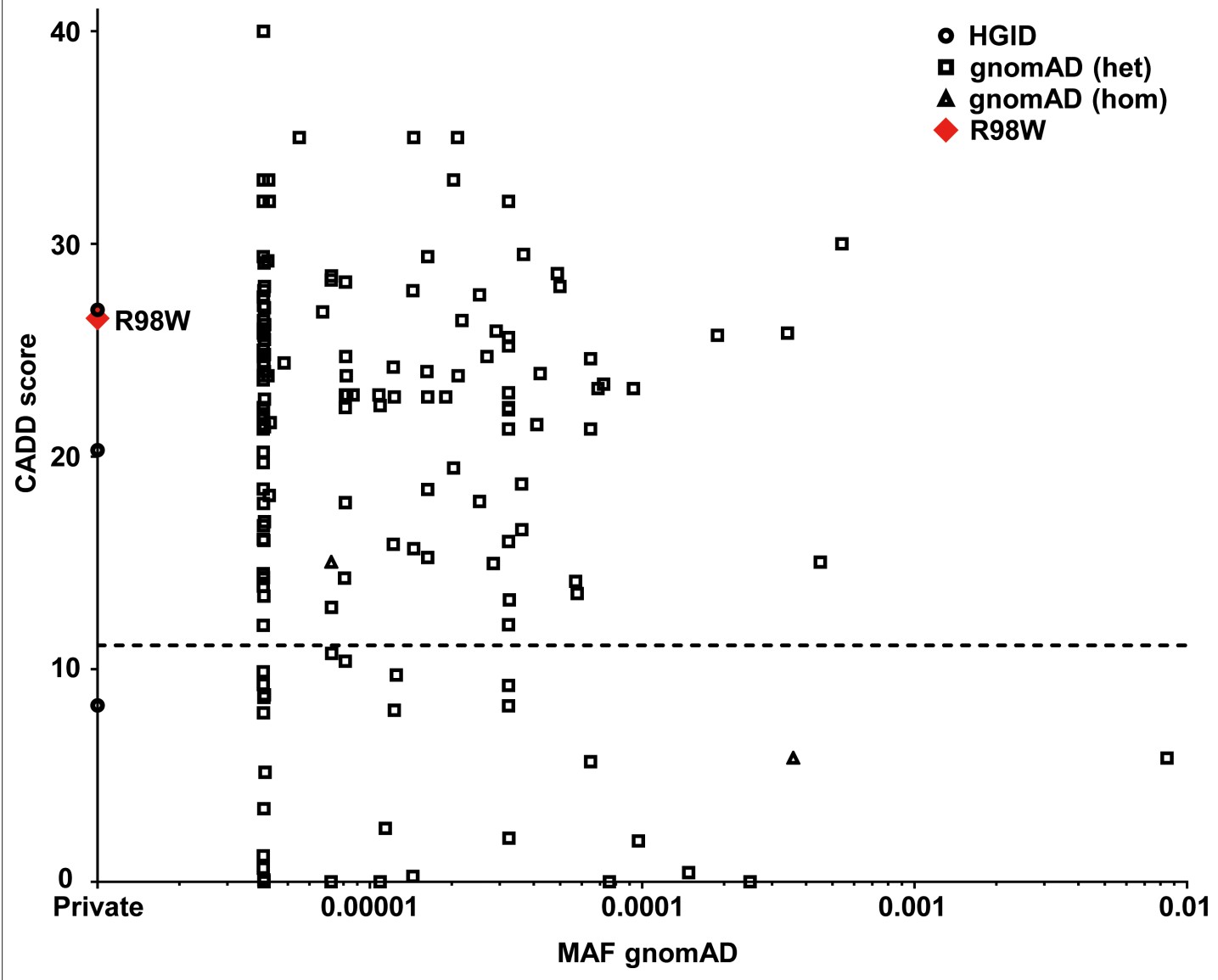

**Figure 2.** Analysis *in silico* of *IRF4* variants. Minor allele frequency (MAF) and combined annotation–dependent depletion (CADD) score of all coding variants previously reported in a public database (gnomAD) (http://gnomad.broadinstitute.org) and in our in-house (HGID) database. The dotted line corresponds to the mutation significance cutoff (MSC) with 95% confidence interval. The R98W variant is shown as a red square.

DOI: https://doi.org/10.7554/eLife.32340.006

The following source data and figure supplements are available for figure 2:

**Source data 1.** 156 non-synonymous heterozygous coding or splice variants reported in the gnomAD or HGID databases.

DOI: https://doi.org/10.7554/eLife.32340.008

**Figure supplement 1.** List of variants and strength of purifying selection on *IRF4*.

DOI: https://doi.org/10.7554/eLife.32340.007

### R98W is predicted to be loss-of-function, unlike most other IRF4 variants

The R98 residue in the DNA-binding domain (DBD) of IRF4 is highly conserved in the 12 species for which *IRF4* has been sequenced (*Figure 1B and D*). It has been suggested that this residue is essential for IRF4 DNA-binding activity, because the R98A-C99A double mutant is loss-of-function (LOF) (*Brass et al., 1999*; *Escalante et al., 2002*). The R98W mutation is predicted to be damaging by multiple programs (*Kircher et al., 2014*); it has a combined annotation–dependent depletion score

(CADD) score of R98W (26.5), well above the mutation significance cutoff (MSC) of *IRF4* (11.125) (*Figure 2*) (*Itan et al., 2016*; *Kircher et al., 2014*). The R98W variant was not present in the gnomAD database or our in-house HGID database of more than 4,000 WES from patients with various infectious diseases. The mutant allele was not found in the sequences for the CEPH-HGDP panel of 1,052 controls from 52 ethnic groups, or in 100 French controls, confirming that this variant was very rare, probably private to this kindred. Therefore, the minor allele frequency (MAF) of this private allele is $<4 \times 10^{-6}$. Moreover, the *IRF4* gene has a gene damage index (GDI) of 2.85, a neutrality index score of 0.15 (*Itan et al., 2015*), and a purifying selection *f* parameter of 0.32 (among the <10% of genes in the genome subject to the greatest constraint; *Figure 2—figure supplement 1*), strongly suggesting that *IRF4* has evolved under purifying selection (i.e. strong evolutionary constraints) (*Eilertson et al., 2012*). Biologically disruptive heterozygous mutations of *IRF4* are therefore likely to have clinical effects. We identified 156 other high-confidence heterozygous non-synonymous coding or splice variants of *IRF4* (*Figure 2—source data 1*) in public (gnomAD: 153 variants, all with MAF <0.009) and HGID (three variants) databases: 147 were missense variants (two of which were also found in the homozygous state: p.S149N and p.A370V), four were frameshift indels leading to premature stop codons (p.W27YfsTer50, p.W74GfsTer28, p.Y152LfsTer60, and p.S160RfsTer11), three were in-frame indels (p.E46del, p.G279_H280del, and p.S435del), one was a nonsense variant (p.R82*), one was an essential splice variant (c.4032T > C), and two were missense variants found only in a non-canonical transcript predicted to undergo nonsense-mediated decay (p.L406P and p.R407W). Up to 150 of the 156 variants are predicted to be benign, whereas only six were predicted to be potentially LOF according to the gnomAD database classification (the four frameshift indels, the nonsense variant, and the essential splice variant). Comparison of the CADD score and MAF of these *IRF4* variants showed R98W to have the second highest CADD score of the four variants with a MAF $<4 \times 10^{-6}$ (*Figure 2*). These findings suggest that the private heterozygous *IRF4* variant of this kindred is biochemically deleterious, unlike most other rare (MAF <0.009) non-synonymous variants in the general population, 150 of 156 of which were predicted to be benign (*Lek et al., 2016*).

## R98W is LOF, unlike most other previously observed IRF4 variants

We first characterized IRF4 R98W production and function *in vitro*, in an overexpression system. We assessed the effect of the *IRF4* R98W mutation on IRF4 levels by transiently expressing WT or mutant R98W in HEK293T cells. IRF4 R98A-C99A, which is LOF for DNA binding (*Brass et al., 1999*), was included as a negative control. In total cell extracts, mutant IRF4 proteins were more abundant than the WT protein and had the expected molecular weight (MW) of 51 kDa, as shown by western blotting (*Figure 3A*). The R98 residue has been shown to be located in a nuclear localization signal, the complete disruption of which results in a loss of IRF4 retention in the nucleus (*Lau et al., 2000*). We therefore analyzed the subcellular distribution of IRF4 WT and R98W proteins, in total, cytoplasmic, and nuclear extracts from transiently transfected HEK293T cells. The R98W mutant was more abundant than the WT protein in total cell and cytoplasmic extracts, but these proteins were similarly abundant in nuclear extracts (*Figure 3B*). We performed luciferase reporter assays to assess the ability of the mutant IRF4 protein to induce transcription from interferon-stimulated response element (*ISRE*) motif-containing promoters. Unlike the WT protein, both R98W and R98A-C99A failed to activate the $(ISRE)_3$ promoter (*Figure 3C*). We also assessed the ability of IRF4 to induce transcription from an (AP-1)-IRF composite element (*AICE*) motif-containing promoter (*Li et al., 2012*). Both R98W and R98A-C99A failed to activate the *AICE* promoter (*Figure 3D*). Moreover, we observed no dominant-negative effect of the IRF4 R98W protein, with either the *ISRE* or *AICE* motif-containing promoter (*Figure 3—figure supplement 1*). We assessed the ability of R98W to bind DNA, in an electrophoretic mobility shift assay (EMSA) (*Figure 3E and F*). Signal specificity was assessed by analyzing both supershift with an IRF4-specific antibody and by competition with an unlabeled competitor probe. The R98W mutation abolished IRF4 binding to the *ISRE cis* element (*Figure 3E*), and binding of the IRF4-PU.1 complex to interferon composite elements (EICEs) containing both IRF4 and PU.1 recognition motifs (*Brass et al., 1999*) (*Figure 3F*). The R98W allele of *IRF4* is therefore LOF for both DNA binding and the induction of transcription. We then tested 153 of the other 156 *IRF4* variants: 150 variants previously described in the gnomAD database and three variants found in the HGID database. The essential splice variant and the two variants present only in a non-canonical transcript were not tested. All the variants tested were normally expressed (two with a higher MW), except for the five predicted LOF variants tested (the

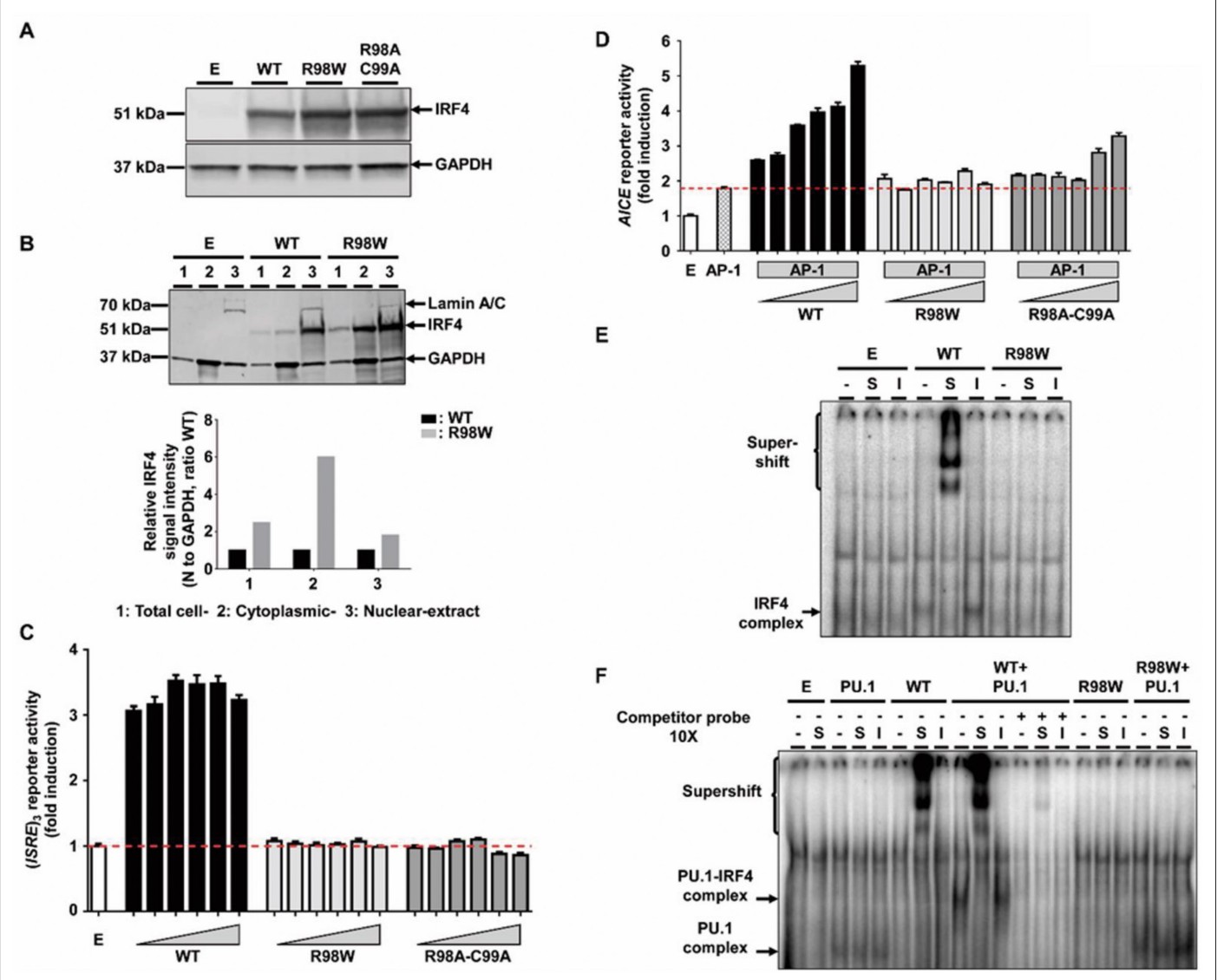

**Figure 3.** Molecular characterization of the R98W *IRF4* mutation (loss of DNA binding). (**A**) HEK293T cells were transfected with the pcDNA3.1 empty vector (**E**) or plasmids encoding *IRF4* WT, *IRF4* R98W or *IRF4* R98A-C99A. Total cell extracts were subjected to western blotting; the upper panel shows IRF4 levels and the lower panel shows the levels of GAPDH, used as a loading control. The results shown are representative of three independent experiments. (**B**) (upper panel) HEK293T cells were transfected with the pcDNA3.1 empty vector (**E**) or plasmids encoding *IRF4* WT or *IRF4* R98W. Total cell (1), cytoplasmic (2) and nuclear (3) extracts were subjected to western blotting. Lamin A/C and GAPDH were used as loading controls. (lower panel) IRF4 signal intensity for R98W-transfected cells and WT-transfected cells, in various cell compartments (total, cytoplasmic and nuclear), normalized against the GAPDH signal, as shown by western blotting. The results shown are representative of three independent experiments. (**C**) Luciferase activity of HEK293T cells cotransfected with an (*ISRE*)$_3$ reporter plasmid plus the pcDNA3.1 empty vector (E, 100 ng) and various amounts of plasmids encoding *IRF4* WT or *IRF4* R98W or *IRF4* R98A/C99A (6.25, 12.5, 25, 50, 75 and 100 ng). Results are shown as the fold induction of activity relative to E-transfected cells. The red dotted line indicates mean activity for E-transfected cells. The mean and standard error of three experiments are shown. (**D**) Luciferase activity of HEK293T cells cotransfected with an *AICE* reporter plasmid plus the pcDNA3.1 empty vector (E, 100 ng) and/ or constant amounts of plasmids encoding *BATF* WT and *JUN* WT (25 ng each, AP-1) and/or various amounts of plasmids encoding *IRF4* WT or *IRF4* R98W or *IRF4* R98A/C99A (6.25, 12.5, 18.8, 25, 37.5 and 50 ng). Results are shown as the fold induction of activity relative to E-transfected cells. The red dotted line indicates mean activity for AP-1-transfected cells. The mean and standard error of two experiments are shown. (**E**) Electrophoretic mobility shift assay (EMSA) with nuclear extracts of HEK293T cells transfected with the pcDNA3.1 empty vector (**E**), or plasmids encoding *IRF4* WT or *IRF4* R98W. Extracts were incubated with a $^{32}$P-labeled *ISRE* probe. Extracts were incubated with a specific anti-IRF4 antibody (**S**) to detect DNA-protein complex supershift, with an isotype control antibody (**I**) to demonstrate the specificity of the complex, and with no antibody (-), as a control. The results shown are representative of three independent experiments. (**F**) EMSA of nuclear extracts of HEK293T cells transfected with the pcDNA3.1 empty vector (**E**), or plasmids encoding *PU.1*, *IRF4* WT, or *IRF4* R98W, or cotransfected with *PU.1* and *IRF4* WT or *PU.1* and *IRF4* R98W plasmids. Extracts were incubated with a $^{32}$P-labeled λB probe (*EICE*). Extracts were incubated with a specific anti-IRF4 antibody (**S**) to detect DNA-protein complex

*Figure 3 continued on next page*

*Figure 3 continued*

supershift, with an isotype control antibody (**I**) to demonstrate the IRF4 specificity of the complex and with no antibody (-), as a control. Experiments in the presence of excess non-radioactive probe (cold probe) demonstrated the probe specificity of the complexes. The results shown are representative of three independent experiments.

DOI: https://doi.org/10.7554/eLife.32340.009

The following figure supplements are available for figure 3:

**Figure supplement 1.** Functional activity of IRF4.

DOI: https://doi.org/10.7554/eLife.32340.010

**Figure supplement 2.** Protein levels of *IRF4* variants previously reported in gnomAD database.

DOI: https://doi.org/10.7554/eLife.32340.011

**Figure supplement 3.** Protein levels of *IRF4* variants from HGID database.

DOI: https://doi.org/10.7554/eLife.32340.012

**Figure supplement 4.** Functional impact of *IRF4* variants previously reported in gnomAD database.

DOI: https://doi.org/10.7554/eLife.32340.013

**Figure supplement 5.** Functional impact of *IRF4* variants from HGID database.

DOI: https://doi.org/10.7554/eLife.32340.014

epitope of the antibody being at the C-terminus of IRF4) (*Figure 3—figure supplements 2; 3*). Transfection with the five predicted LOF plasmids was efficient, as assessed by cDNA amplification and sequencing (data not shown). When tested for $(ISRE)_3$ promoter activation, the five stop-gain or frameshift variants predicted to be LOF in the gnomAD database (very rare variants, MAF $<6 \times 10^{-6}$) were found to be LOF. We also showed that among all the non-synonymous coding variants tested, only one rare very rare (MAF $9 \times 10^{-6}$) inframe deletion of one amino acid (E46del) reported in the gnomAD database, was hypomorphic, and another variant from our in-house WES database (G279_H280 del, private to one family) was LOF (*Figure 3—figure supplements 4; 5*). The cumulative frequency of these seven LOF ($n = 6$) or hypomorphic ($n = 1$) variants was $<4 \times 10^{-5}$, fully consistent with the frequency of WD (occurring only in adults chronically infected with Tw). Overall, our data show that the R98W *IRF4* allele is LOF, like only six other very rare non-synonymous *IRF4* coding variants of the 153 variants tested. Moreover, R98W is not dominant negative.

## AD IRF4 deficiency phenotypes in heterozygous EBV-B cells

We investigated the cellular phenotype of heterozygosity for the R98W allele in EBV-transformed B-cell lines (EBV-B cells) from patients. We performed reverse transcription-quantitative polymerase chain reaction (RT-qPCR) on EBV-B cells from P1, P3, two healthy heterozygous relatives (*IRF4* WT/R98W), four healthy *IRF4*-WT homozygous relatives, and seven healthy unrelated controls. We also investigated 25 unrelated WD patients with Tw carriage. We sequenced all *IRF4* coding exons for these patients, who were found to be WT. They were also found to have an intact *IRF4* cDNA structure and normal IRF4 protein levels in EBV-B cells (data not shown). Cells from individuals heterozygous for the R98W mutation (patients and healthy carriers) had higher *IRF4* mRNA levels than those from WT homozygous relatives, unrelated WD cohort patients and EBV-B cells from healthy unrelated controls (*Figure 4A*). We compared the relative abundances of WT and R98W *IRF4* mRNA in EBV-B cells from heterozygous carriers of the mutation, by performing TA-cloning experiments on P1, P3, one healthy heterozygous relative, one relative homozygous for WT *IRF4*, and two previously tested healthy unrelated controls. In heterozygous carriers of the mutation (patients and healthy relatives), the R98W mutation was present in 48.1–60% of the total *IRF4* mRNA, whereas the rest was WT (*Figure 4B*). We evaluated the levels and distribution of IRF4 protein by western blotting on EBV-B cells from P1, P2, P3, one healthy heterozygous relative, three healthy homozygous WT relatives and five unrelated healthy individuals. As in transfected HEK293T cells, IRF4 protein levels were high both in total cell extract and even more so in cytoplasmic extracts of EBV-B cells from heterozygous carriers (*Figure 4—figure supplement 1*). By contrast, IRF4 protein levels in EBV-B cell nuclei were similar in heterozygous carriers and controls (*Figure 4—figure supplement 1*). As IRF4 is a transcription factor, we then analyzed the steady-state transcriptome of EBV-B cells from three healthy homozygous WT

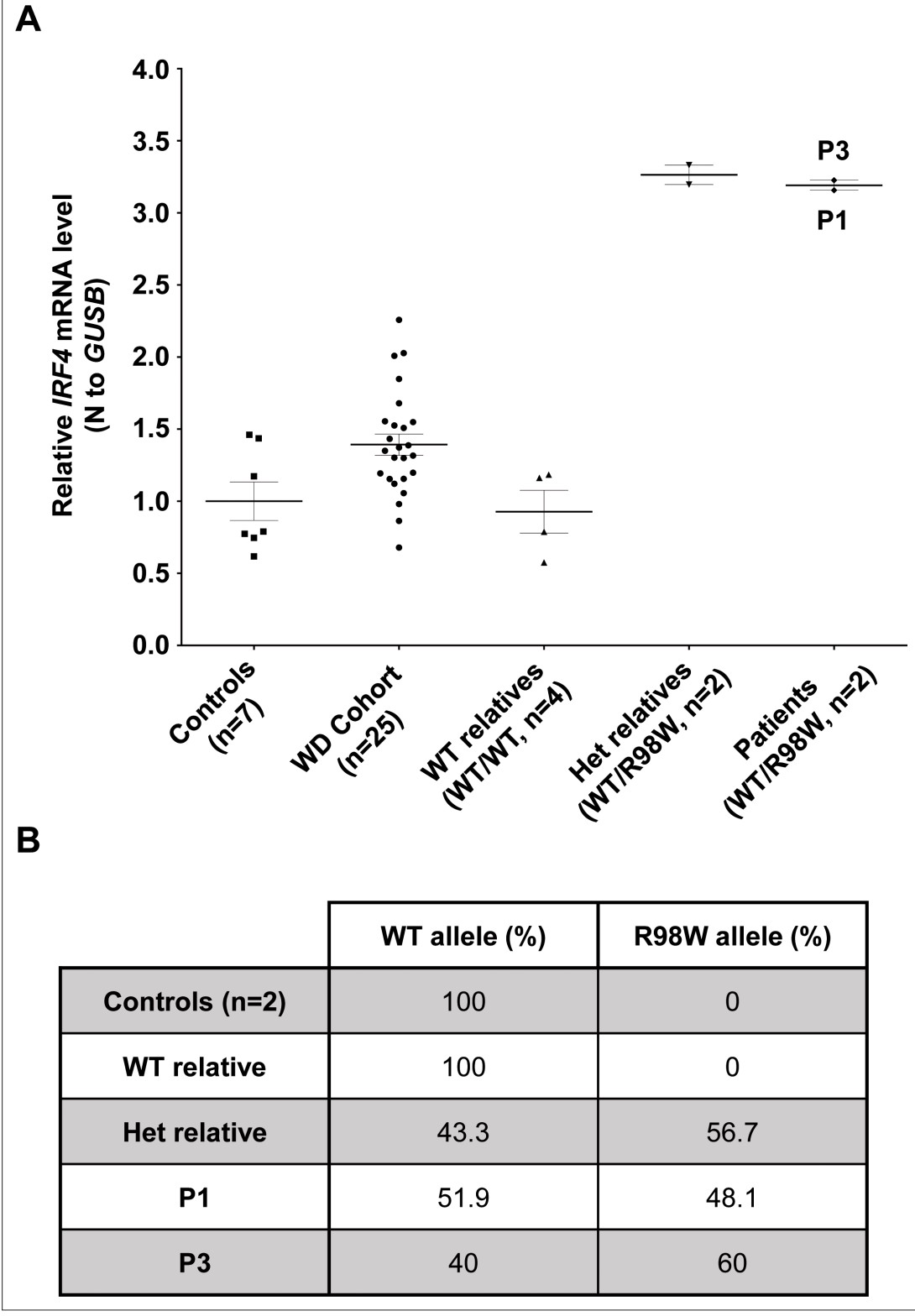

**Figure 4.** *IRF4* mRNA levels in EBV-B cells.  (**A**) Total RNA extracted from healthy unrelated controls (*n* = 7; *IRF4* WT/WT), patients diagnosed with Whipple's disease (*n* = 25; WT/WT for all coding exons of *IRF4*) not related to this kindred, healthy homozygous WT relatives (*n* = 4, *IRF4* WT/WT), patients with monoallelic *IRF4* mutations (*n* = 2; *IRF4* WT/R98W) and asymptomatic heterozygous relatives with monoallelic *IRF4* mutations (*n* = 2; *IRF4* WT/R98W) was subjected to RT-qPCR for total *IRF4*. Data are displayed as 2-ΔΔCt after normalization according to endogenous *GUSB* control gene expression (ΔCt) and the mean of controls (ΔΔCt). The results shown are the mean ± SD of three independent experiments. (**B**) Calculated frequency

*Figure 4 continued on next page*

*Figure 4 continued*

(%) of each mRNA (WT and R98W allele) obtained by the TA-cloning of cDNA generated from EBV-B cells from healthy unrelated controls (n = 2), healthy homozygous WT relatives (n = 1), patients with monoallelic *IRF4* mutations (n = 2) and asymptomatic heterozygous relatives with monoallelic *IRF4* mutations (n = 1).

DOI: https://doi.org/10.7554/eLife.32340.015

The following figure supplement is available for figure 4:

**Figure supplement 1.** IRF4 protein levels in EBV-B cells.

DOI: https://doi.org/10.7554/eLife.32340.016

relatives and three WT/R98W heterozygotes (P1, P3, VI.6). We identified 37 protein-coding genes as differentially expressed between subjects heterozygous for *IRF4* and those homozygous WT for *IRF4* (18 upregulated and 19 downregulated; data not shown). We identified no marked pathway enrichment based on these genes. EBV-B cells from individuals heterozygous for *IRF4* had a detectable phenotype, in terms of IRF4 production and function, consistent with AD IRF4 deficiency underlying WD.

## AD IRF4 deficiency phenotypes in heterozygous leukocytes

We assessed IRF4 levels in peripheral blood mononuclear cells (PBMCs) from healthy controls (*Figure 5—figure supplement 1*). IRF4 were also expressed in CD4$^+$ T cells, particularly after stimulation with activating anti-CD2/CD3/CD28 monoclonal antibody-coated (mAb-coated) beads (data not shown). We therefore assessed the IRF4 protein expression profile in CD4$^+$ T cells from four healthy unrelated controls, P1 and P3, with and without (non-stimulated, NS) stimulation with activating anti-CD2/CD3/CD28 mAb-coated beads. The results were consistent with those for transfected HEK293T and EBV-B cells, as IRF4 levels were higher in activated CD4$^+$ T cells from P1 and P3 than in controls, both for total cell extracts, and even more so for the cytoplasmic compartment (*Figure 5A and B*). By contrast, IRF4 levels in the nucleus were similar and, possibly, even slightly lower in patients than in controls (*Figure 5C*). We also investigated peripheral myeloid (*Figure 5—figure supplements 1–4*) and lymphoid blood cell subsets in patients (*Figure 5—figure supplements 5–7*; *Figure 5—source data 1*), which display an apparently normal development compared to healthy controls' cells. Then, we checked for transcriptomic differences associated with genotype and/or infection, by investigating the transcriptomes of PBMCs from six *IRF4*-heterozygous individuals (three patients, P1-P3; and three healthy relatives, HET1-HET3) and six *IRF4* WT-homozygous individuals (four healthy relatives, WT1-WT4; and two healthy unrelated controls, C1-C2) with and without *in vitro* infection for 24 hr with Tw, or *Mycobacterium bovis*-Bacillus Calmette-Guerin (BCG), which, like Tw, belongs to phylum Actinobacteria. We performed unsupervised hierarchical clustering of the differentially expressed (DE) transcripts (infected versus uninfected) to analyze the overall responsiveness of PBMCs from individual subjects to BCG and Tw infections *in vitro*. Heterozygous individuals clearly clustered separately from homozygous WT individuals (*Figure 6A*), revealing a correlation between genotype and response to infection. Overall, we found that 402 transcripts from 193 unique genes were responsive to BCG infection (*Figure 6—source data 1*), and 119 transcripts from 29 unique genes were responsive to Tw infection (*Figure 6—source data 2*) in homozygous WT subjects, according to the criteria described in the Materials and methods. Due to the small number of Tw-responsive transcripts linked to unique genes, we were unable to detect any pathway enrichment for this specific condition. However, we identified 24 canonical pathways as enriched after the exposure of PBMCs to BCG. We ranked these pathways according to the difference in mean z-score between homozygous WT and heterozygous subjects (*Figure 6B*). The top 10 pathways included the interferon signaling network, the Th1 pathway network, the HMGB1 signaling network, the p38 MAPK signaling network, the NF-κB signaling network, the dendritic cell maturation network and the network responsible for producing nitric oxide and reactive oxygen species. These pathways were highly ranked mostly due to *IFNG* and *STAT1*, which were strongly downregulated in *IRF4* heterozygotes, particularly in P1, P2 and P3, relative to WT homozygotes. IRF4 is predicted to bind the promoter regions of 47% of the genes identified in the BCG study (91 of 193 genes), including those of *IFNG* and *STAT1*. Subjects heterozygous for *IRF4* also had lower levels of *LTA* expression, and lower levels of *IL2RA* expression were observed specifically in patients (GSE102862). These data suggest a general impairment of the

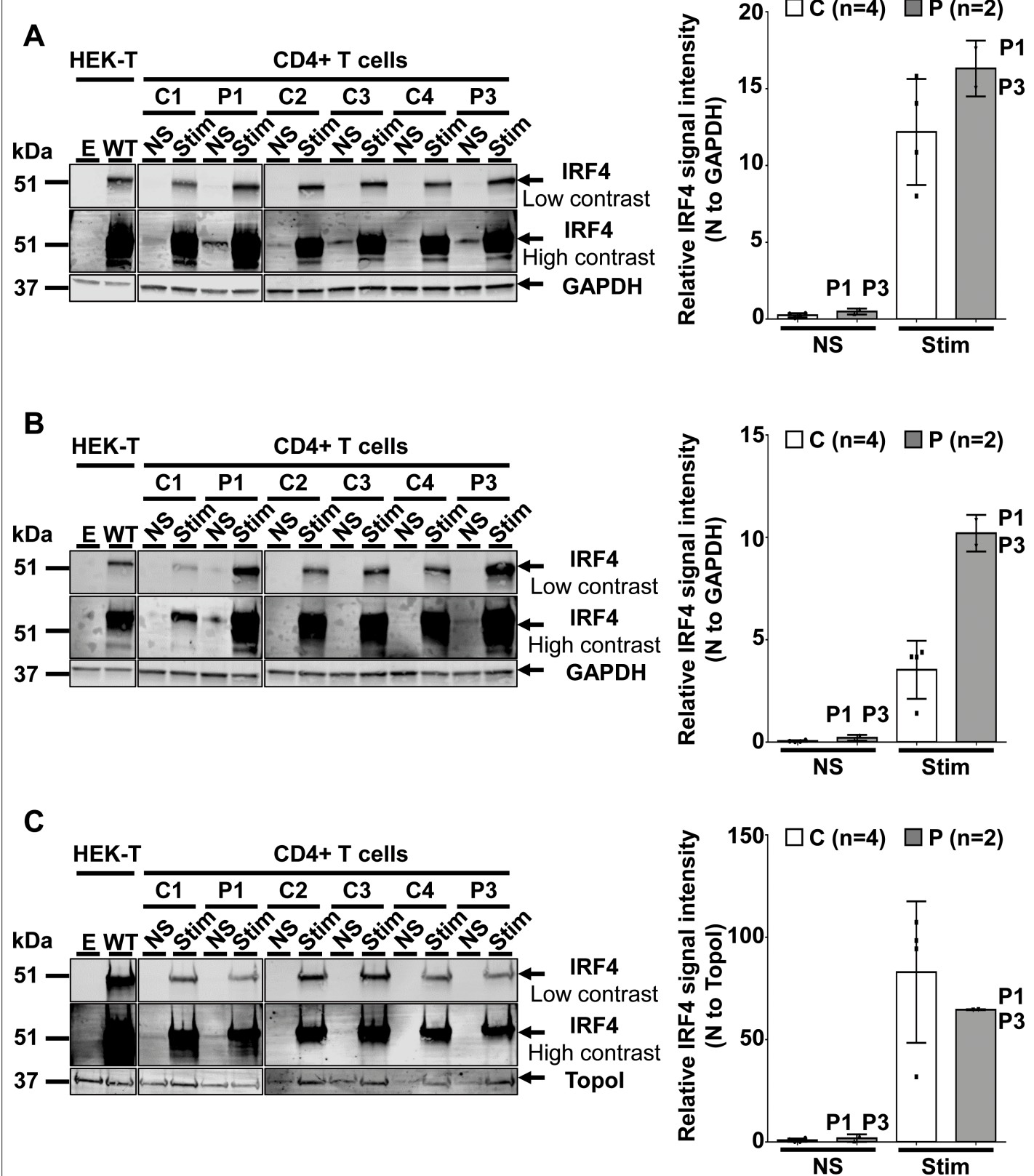

**Figure 5.** IRF4 protein levels in CD4+ T cells. (**A–C**) (Left) Total-cell (**A**), cytoplasmic (**B**) and nuclear (**C**) extracts from CD4+ T cells from four healthy unrelated controls (C1 to C4) and two patients (P1 and P3) stimulated with activating anti-CD2/CD3/CD28 monoclonal antibody-coated beads (Stim)

*Figure 5 continued on next page*

*Figure 5 continued*

or left unstimulated (NS). Protein extracts from HEK293T cells transfected with the pcDNA3.1 empty vector (**E**) or plasmids encoding *IRF4* WT plasmids were used as controls for the specific band corresponding to IRF4. (Right) Representation of IRF4 signal intensity for each individual, obtained by western blotting, with normalization against the GAPDH signal (total, cytoplasmic extracts) or the topoisomerase I signal (nuclear extracts).

DOI: https://doi.org/10.7554/eLife.32340.017

The following source data and figure supplements are available for figure 5:

**Source data 1.** Immunophenotyping of patients (P1, P2 and P3) and a WT homozygous relative.

DOI: https://doi.org/10.7554/eLife.32340.029

**Figure supplement 1.** IRF4 protein levels in PBMC subpopulations.

DOI: https://doi.org/10.7554/eLife.32340.018

**Figure supplement 2.** Percentage of dendritic cells and monocyte subtypes within total PBMCs.

DOI: https://doi.org/10.7554/eLife.32340.019

**Figure supplement 3.** IRF4 levels in controls and patient monocyte-derived macrophages.

DOI: https://doi.org/10.7554/eLife.32340.020

**Figure supplement 4.** Surface marker levels in controls and patient monocyte-derived macrophages.

DOI: https://doi.org/10.7554/eLife.32340.021

**Figure supplement 5.** Percentage of memory B cells in PBMCs from controls and patients.

DOI: https://doi.org/10.7554/eLife.32340.022

**Figure supplement 6.** *In vitro* differentiation of CD4[+] T cells from patients and controls.

DOI: https://doi.org/10.7554/eLife.32340.023

**Figure supplement 7.** *Ex vivo* cytokine production by CD4[+] memory T cells from patients and controls.

DOI: https://doi.org/10.7554/eLife.32340.024

T-cell response in subjects heterozygous for *IRF4* upon BCG infection *in vitro*. Moreover, the lower levels of *CD80* expression suggest a possible impairment of myeloid and/or antigen-presenting cell function upon BCG infection in patients, but not in healthy heterozygous or homozygous WT subjects (GSE102862). Peripheral leukocytes from *IRF4*-heterozygous individuals therefore had a phenotype in terms of IRF4 production and function.

## Discussion

WD was initially described as an inflammatory disease (*Whipple, 1907*) but was subsequently shown to be infectious (*Raoult et al., 2000*; *Relman et al., 1992*; *Yardley and Hendrix, 1961*). We provide evidence that WD is also a genetic disorder. We show here that, in a large multiplex kindred, heterozygosity for the private, LOF R98W mutation of *IRF4* underlies an AD form of WD with incomplete penetrance. The causal relationship between *IRF4* genotype and WD was demonstrated as follows. First, the *IRF4* R98W mutation is the only non-synonymous rare variant segregating with WD in this kindred. Second, the mutation was demonstrated experimentally to be LOF, unlike 146 of 153 other non-synonymous coding *IRF4* variants in the general population. Only seven of the 153 non-synonymous coding variants identified (including the five predicted to be LOF in the gnomAD database) were found to be LOF ($n = 6$) or hypomorphic ($n = 1$) and they were all extremely rare (cumulative MAF $<4 \times 10^{-5}$). Moreover, *IRF4* has evolved under purifying selection, suggesting that deleterious heterozygous variants of this gene entail fitness costs (*Barreiro and Quintana-Murci, 2010*; *Quintana-Murci and Clark, 2013*; *Rieux-Laucat and Casanova, 2014*). Third, EBV-B cells and activated CD4[+] T cells heterozygous for *IRF4* R98W have a distinctive phenotype, particularly for IRF4 expression in the cytoplasm. This mutation also has a strong functional impact on gene expression in *IRF4* R98W-heterozygous PBMCs stimulated with BCG or Tw. These findings unequivocally show that heterozygosity for the R98W allele of *IRF4* is the genetic etiology of WD in this kindred. Although we did not find any *IRF4* mutations in a pilot cohort of 25 patients with sporadic WD, these and other patients may also develop WD due to other inborn errors of immunity, possibly related to IRF4, as suggested by the apparent genetic heterogeneity and physiological homogeneity underlying severe infectious diseases (*Andersen et al., 2015*; *Casanova, 2015a*; *Casanova, 2015b*; *Ciancanelli et al., 2015*; *Israel et al.,*

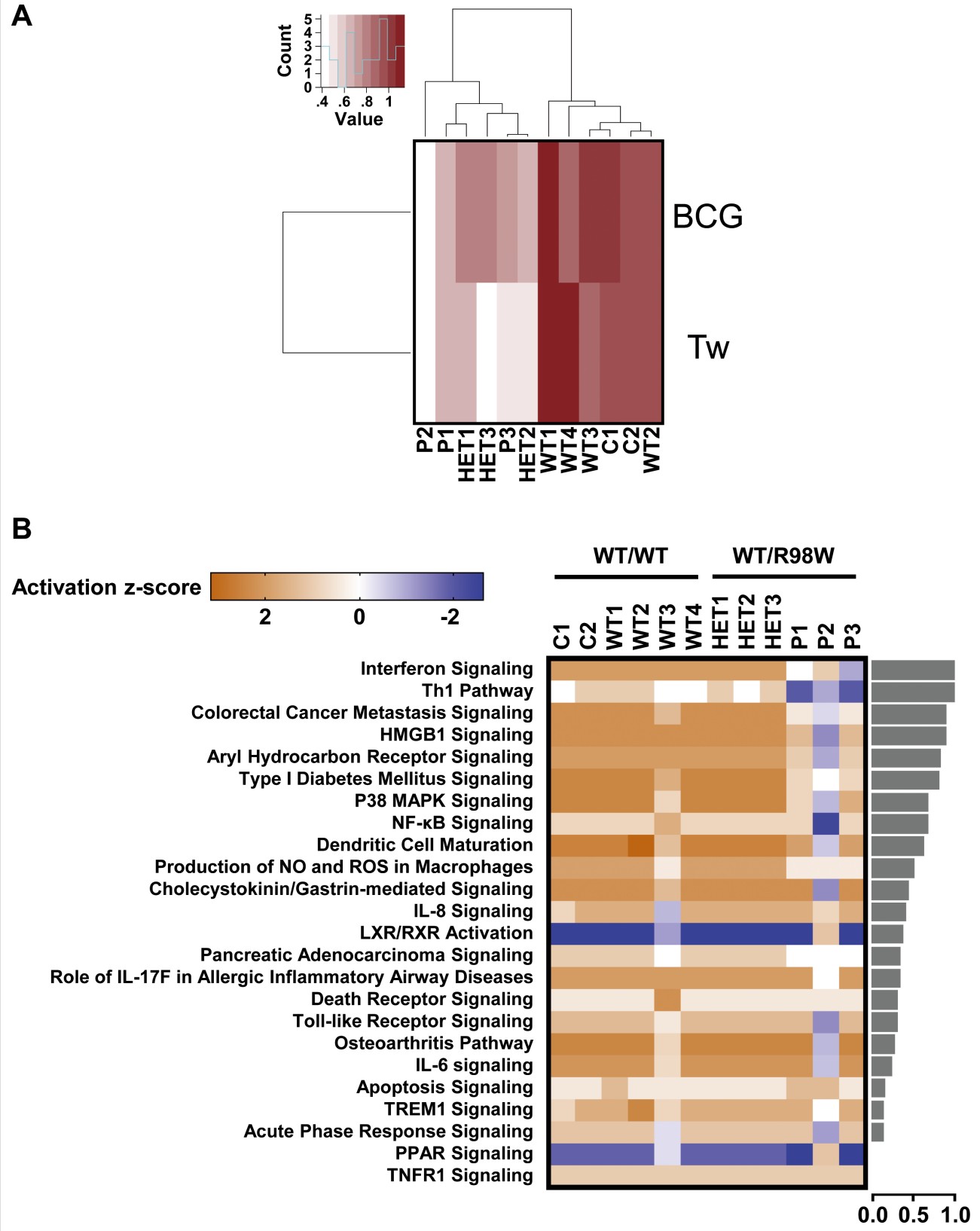

**Figure 6.** Overall transcriptional responsiveness of PBMCs following *in vitro* exposure to Tw and BCG and pathway activity analysis for genes responsive to BCG exposure. (**A**) The overall responsiveness of individual subjects following stimulation with BCG and Tw, relative to non-stimulated

*Figure 6 continued on next page*

*Figure 6 continued*

conditions (along the horizontal axis) is shown as a heatmap. For each individual and each stimulus, overall responsiveness was assessed on the basis of normalized counts of differentially expressed transcripts, as described in the corresponding Materials and methods section. Subjects were grouped by unsupervised hierarchical clustering. (B) Enriched canonical pathways were ranked according to differences in mean activation z-score between genotypes (WT/WT individuals vs. WT/R98W individuals). The activation z-scores for each individual and pathway are shown as heat maps. Pathways predicted to be activated are depicted in orange, pathways predicted to be inhibited are depicted in blue. A lack of prediction concerning activation is depicted in white. Individuals are presented in columns, pathways in rows. The pathways are ranked from most different between genotypes (at the top of the list) to the least different (at the bottom). The differences in mean activation z-scores between WT/WT and WT/R98W individuals for each pathway are depicted as bars to the right of the heat maps (the direction of difference is not shown). The Ingenuity Pathway Analysis (IPA) tool was used to generate a list of the most significant canonical pathways and their respective activation z-scores.

DOI: https://doi.org/10.7554/eLife.32340.026

**Source data 1.** Differentially expressed (DE) genes found to be responsive to BCG in homozygous WT subjects using the criteria described in the Materials and methods section.
DOI: https://doi.org/10.7554/eLife.32340.027

**Source data 2.** Differentially expressed (DE) genes found to be responsive to Tw in homozygous WT subjects using the criteria described in the Materials and methods section.
DOI: https://doi.org/10.7554/eLife.32340.028

*2017*; *Lamborn et al., 2017*; *Ogunjimi et al., 2017*; *Vanhollebeke et al., 2006*; *Zhang et al., 2018*; *Zhang et al., 2015*). This observation therefore extends our model, in which life-threatening infectious diseases striking otherwise healthy individuals during primary infection can result from single-gene inborn errors of immunity.

In this kindred with AD IRF4 deficiency, haploinsufficiency was identified as the key mechanism, although IRF4 protein levels in the cytoplasmic compartment were higher in patients with the mutation than in wild-type homozygotes. The protein was not more abundant in the nucleus, where IRF4 exerts its effects on transcription. Moreover, half of the *IRF4* mRNA in EBV-B cells from heterozygous subjects is WT. The total amount of *IRF4* mRNA was higher in the EBV-B cells of heterozygous subjects, but the total amount of IRF4 protein in the nuclear compartment of heterozygous EBV-B and activated CD4$^+$ T cells was similar to that in WT homozygous cells. These data suggest that no more than half the IRF4 protein in the nucleus is WT in heterozygous cells. In addition, not only is *IRF4* subject to purifying selection, but also the R98W mutation is itself LOF, with no detectable dominant-negative effect at cell level. Haploinsufficiency is an increasingly recognized mechanism underlying AD inborn errors of immunity (*Afzali et al., 2017*; *Rieux-Laucat and Casanova, 2014*). It is commonly due to loss-of-expression alleles, contrasting with the negative dominance typically exerted by expressed proteins, but many mutations are known to cause haploinsufficiency without actually preventing protein production (*Afzali et al., 2017*; *Pérez de Diego et al., 2010*; *Rieux-Laucat and Casanova, 2014*). Haploinsufficiency in this kindred is not due to loss-of-expression of IRF4. Instead, it results from a lack of activity of the R98W IRF4 proteins present in the nucleus.

Incomplete penetrance is common in conditions resulting from haploinsufficiency. In this kindred, incomplete penetrance may result from a lack of Tw infection (in heterozygous individuals IV.5 and VI.7), or a lack of WD development in infected individuals (in heterozygous individuals III.6, IV.4, V.3, V.4, VI.6). All five chronic carriers of Tw were heterozygous for the *IRF4* R98W mutation, suggesting that AD IRF4 deficiency also favors the development of chronic Tw carriage. The five asymptomatic carriers were 24 to 82 years old, whereas the four patients were 69 to 92 years old. The impact of IRF4 R98W may therefore increase with age, initially facilitating chronic carriage in Tw-infected individuals, and subsequently predisposing chronic carriers to the development of WD. We cannot exclude the possibility that a modifier allele at another locus contributes to the development of WD in infected heterozygous individuals with *IRF4* mutations. Future studies will attempt to define the cellular basis of WD in individuals with *IRF4* mutations. The apparently normal development of all peripheral myeloid and lymphoid blood cell subsets studied in patients, and the selective predisposition of these individuals to WD suggest that the disease mechanism is subtle and specifically affects protective immunity to Tw and that it may act in the gastrointestinal (GI) tract. Interestingly, the data of several public databases indicate that *IRF4* RNA is expressed in the stomach, colon, and small intestine (https://www.gtexportal.org, http://biogps.org), and that the IRF4 protein is expressed in glandular cells from the stomach, duodenum, small intestine, and rectum (https://www.proteinatlas.org). In addition, a

recent analysis of human intestinal macrophage subsets (*Bujko et al., 2018*) showed *IRF4* RNA to be expressed in human intestinal myeloid resident cells. Further studies of GI-tract-resident cells, including myeloid and lymphoid cells in particular, should make it possible to decipher the molecular and cellular mechanisms by which human IRF4 haploinsufficiency underlies WD upon infection by Tw.

# Materials and methods

**Key resources table**

| Reagent type (species) or resource | Designation | Source or reference | Identifiers | Additional information |
|---|---|---|---|---|
| Gene (Human) | IRF4 (NM_002460.3) | This paper | | Vector backbone: pcDNA 3.1D/V5-His-TOPO vector (Thermo Fisher Scientific) |
| Gene (Human) | PU.1 (NM_001080547.1) | This paper | | Vector backbone: pcDNA 3.1D/V5-His-TOPO vector (Thermo Fisher Scientific) |
| Gene (Human) | BATF (NM_006399.3) | OriGene | RC207104 | |
| Gene (Human) | JUN (NM_002228.3) | OriGene | RC209804 | |
| Strain (Tropheryma whipplei), strain background (DIG APD 25) | Tw | This paper | NCBI taxon: 2039 | Obtained from 'Research Unit of Infectious and Tropical Emerging Diseases, University Aix-Marseille, URMITE, UM63, CNRS 7278, IRD 198, 13005 Marseille, France, EU'. Strain isolated from mesenteric lymph node (29/01/09). |
| Strain (Mycobacterium bovis-Bacillus Calmette-Guerin) , strain background (pasteur) | BCG | doi: 10.1084/jem.20021769 | NCBI taxon: 33892 | |
| Cell line (Human) | HEK293T | ATCC | CRL-3216 | |
| Cell line (Human) | EBV-B cells | This paper | | For each individual, purified B cells were immortalized with EBV in the laboratory |
| Transfected construct (PGL4.10[luc2]) | (ISRE)3 reporter plasmid, | This paper, backbone: Promega | #E6651 | Obtained from 'Department of Biotechnology and Food Engineering, Technion-Israel Institute of Technology' |
| Transfected construct (PGL4.10[luc2]) | AICE reporter plasmid | This paper, backbone: Promega | #E6651 | Generated by metabion international ag |
| Transfected construct (pRL-SV40 vector) | pRL-SV40 vector | Promega | #E2231 | |
| Biological sample (Human) | Patients' blood samples | This paper | | |
| Biological sample (Human) | Controls' blood samples | This paper | | |
| Antibody | anti-IRF4 | Santa Cruz | M-17 | Dilution: 1/1000 |
| Antibody | anti-GAPDH | Santa Cruz | FL-335 | Dilution: 1/1000 |
| Antibody | anti-topoisomerase I | Santa Cruz | C-21 | Dilution: 1/1000 |
| Antibody | anti-lamin A/C | Santa Cruz | H-110 | Dilution: 1/1000 |
| Antibody | anti-CD11b | Miltenyi Biotec | # 130-110-611 | Fluorochrome: PE |
| Antibody | anti-CD86 | Miltenyi Biotec | #130-094-877 | Fluorochrome: PE |
| Antibody | anti-CD206 | Miltenyi Biotec | #130-099-732 | Fluorochrome: PE |
| Antibody | anti-CD209 | Miltenyi Biotec | #130-109-589 | Fluorochrome: PE |

| Reagent type (species) or resource | Designation | Source or reference | Identifiers | Additional information |
|---|---|---|---|---|
| Antibody | anti-HLADR | Miltenyi Biotec | #130-111-789 | Fluorochrome: PE |
| Antibody | anti-CD20 | BD biosciences | | Fluorochrome: PE; clone H1 |
| Antibody | anti-CD10 | BD biosciences | | Fluorochrome: APC, clone HI10a |
| Antibody | anti-CD27 | BD biosciences | | Fluorochrome: PerCP-Cy5.5; clone L128 |
| Antibody | anti-IgM | Miltenyi | | Clone PJ2-22H3 |
| Antibody | anti-IgG | BD biosciences | | Fluorochrome: BV605; clone G18-145 |
| Antibody | anti-IgA | Miltenyi | | Clone IS11-8E10 |
| Antibody | anti-CD4 | eBioscience | | Fluorochrome: Pacific blue; clone OKT4 |
| Antibody | anti-CD45RA | BD biosciences | | Fluorochrome: PerCP-Cy5.5, clone HI100 |
| Antibody | anti-CCR7 | Sony | | Fluorochrome: FITC; clone G043H7 |
| Recombinant DNA reagent | pcDNA 3.1D/V5-His-TOPO vector | Thermo Fisher Scientific | #K4900-01 | |
| Sequence-based reagent | IRF4-specific primer | Thermo Fisher Scientific | #Hs01056533_m1 | |
| Sequence-based reagent | GUSB | Thermo Fisher Scientific | #4326320E | |
| Peptide, recombinant protein | rhGM-CSF | R and D System | #CAA26822 | |
| Peptide, recombinant protein | rhM-CSF | R and D System | #NP_757350 | |
| Peptide, recombinant protein | IFN- | Boehringer Ingelheim | | Imukin |
| Peptide, recombinant protein | rhIL4 | R and D System | #P05112 | |
| Commercial assay or kit | Lipofectamine LTX kit | Thermo Fisher Scientific | #15338100 | |
| Commercial assay or kit | Dual-Luciferase 1000 assay system kit | Promega | #E1980 | |
| Commercial assay or kit | ZR RNA Microprep kit | Zymo research | #R1061 | |
| Commercial assay or kit | High-Capacity RNA-to-cDNA kit | Thermo Fisher Scientific | #R4387406 | |
| Commercial assay or kit | TOPO TA cloning kit | Thermo Fisher Scientific | #K450001 | |
| Commercial assay or kit | directional TOPO expression kit | Thermo Fisher Scientific | #K4900-01 | |
| Commercial assay or kit | QuikChangeII XL Site-Directed Mutagenesis Kit | Agilent Technologies | #200522 | |
| Commercial assay or kit | LIVE/DEAD Fixable Aqua Dead Cell Stain Kit | Thermo Fisher Scientific | #L34957 | |
| Chemical compound, drug | Tris | MP biomedicals | #11TRIS01KG | |
| Chemical compound, drug | HCl | Sigma | #H1758 | |
| Chemical compound, drug | NaCl | Sigma | #S3014 | |
| Chemical compound, drug | Triton X-100 | Sigma | #T8532 | |
| Chemical compound, drug | EDTA | MP biomedicals | #11EDTA05M1 | |
| Chemical compound, drug | protease inhibitors Complete | Roche | #04693116001 | |
| Chemical compound, drug | Phosphatase inhibitor cocktail | Roche | #04906837001 | |

| Reagent type (species) or resource | Designation | Source or reference | Identifiers | Additional information |
|---|---|---|---|---|
| Chemical compound, drug | DTT | Thermo Fisher Scientific | #20290 | |
| Chemical compound, drug | pepstatin A | Sigma | #P4265 | |
| Chemical compound, drug | leupeptin | Sigma | #L2884 | |
| Chemical compound, drug | antipain | Sigma | #A6191 | |
| Chemical compound, drug | Hepes | Sigma | #H3375 | |
| Chemical compound, drug | KCl | Sigma | #P9333 | |
| Chemical compound, drug | EGTA | Amresco | #0732 | |
| Chemical compound, drug | NP40 | Sigma | #N6507 | |
| Chemical compound, drug | NaF | Sigma | #S7920 | |
| Chemical compound, drug | PMSF | Sigma | #P7626 | |
| Chemical compound, drug | MgCl2 | Sigma | #M8266 | |
| Chemical compound, drug | Klenow fragment | NEB | #M0210S | |
| Chemical compound, drug | d-ATP-32P | PerkinElmer | #BLU012H250UC | |
| Chemical compound, drug | TBE migration buffer | Euromedex | #ET020-B | |
| Chemical compound, drug | acrylamide/bis-acrylamide 37.5:1 | Sigma | #A7168 | |
| Software, algorithm | affy R package | Gautier et al., 2004; Irizarry et al., 2003 | | |
| Software, algorithm | IPA software | Alsina et al., 2014 | | |
| Software, algorithm | Microsoft Excel | Microsoft | | |
| Software, algorithm | GraphPad Prism V7.0 | GraphPad | | |
| Software, algorithm | Image studio | Licor | | |

## Patients and family

All members of the multiplex kindred studied, the pedigree of which is shown in *Figure 1A*, live in France and are of French descent. Informed consent was obtained from all family members, and the study was approved by the national ethics committee.

Patient 1 (P1, proband) was born in 1948 and presented arthritis of the right knee in 2011, after recurrent episodes of arthritis of this joint associated with effusion since 1980. *Tropheryma whipplei* (Tw) was detected in synovial fluid by PCR and culture in 2011, but was not detected by PCR in saliva, feces, and small-bowel biopsy specimens. Physical examination revealed a large effusion of the right knee, limiting mobility. The fluid aspirated from this joint contained 4,000 erythrocytes/mm$^3$ and 8,800 leukocytes/mm$^3$, but no crystals or evidence of microbes. Synovial hypertrophy of the right knee and a narrowing of the right internal femoro-tibial joint were detected on MRI. X-ray showed an extension of the right femoro-tibial joint and erosion of the posterior part of the femoro-tibial joint. However, erythrocyte sedimentation rate (ESR) (3 mm/h) and C-reactive protein (CRP) (1.8 mg/l) determinations gave negative results. P1 received methotrexate (15 mg/week) for 4 months, without remission. Antibiotic treatment with doxycycline (200 mg/day) was then immediately initiated. The arthralgia resolved, but right knee effusion persisted. Hydroxychloroquine was therefore added to the treatment regimen. At last follow-up, in 2016, the patient was well.

P2, a second cousin of P1, was born in 1941 and was diagnosed with classical WD and digestive problems in 1978, based on positive periodic acid–Schiff (PAS) staining of a small intestine biopsy specimen. She was treated with sulfamethoxazole/trimethoprim. At last follow-up, in 2016, Tw PCR was positive for the saliva and feces.

P3, the father of P1, was born in 1925 and was diagnosed with classical WD in 1987 on the basis of positive PAS staining of a small intestine biopsy specimen. Clinical manifestations included diarrhea, abdominal pain and weight loss. P3 displayed no extraintestinal manifestations. He was successfully treated with sulfamethoxazole/trimethoprim, with complete clinical and bacteriological remission.

P4, the brother of P2, was born in 1947 and sought medical advice in 2015 for arthralgia affecting the knees and right ulna-carpal joints. The other joints were unaffected. A culture of the joint fluid was negative for bacteria, but Tw was not sought. Tw was not detected in the saliva and feces by PCR or culture, but serological tests for Tw were positive. The fluid aspirated from the right knee contained 4,800 erythrocytes/mm$^3$ and 10,900 leukocytes/mm$^3$ (91% neutrophils and 9% lymphocytes) without crystals. Blood tests revealed an ESR of 30 mm/h and a CRP concentration of 50 mg/l, with no rheumatoid factor, anti-cyclic citrullinated peptide antibodies (anti-CCP) or anti-nuclear antibodies. An X-ray revealed a narrowing of the joint space in the knees and vertebral hyperostosis was visible. The joints of the hands were unaffected. The patient was treated with anti-inflammatory drugs, without success. Treatment with methotrexate and steroids was introduced, followed by antibiotics, the effect of which is currently being evaluated.

Saliva and/or feces samples from 18 other members of the family were checked for the presence of Tw, by a PCR specifically targeting *T. whipplei,* as previously described (*Figure 1A*, *Figure 1—source data 1*) (*Edouard et al., 2012*). Five individuals were found to be chronic carriers (mean age: 55 years) and 13 were not (mean age: 38 years). Testing was not possible for nine other relatives. The overall distribution of WD in this kindred was suggestive of an AD trait with incomplete penetrance.

## Genome-wide analysis

Genome-wide linkage analysis was performed by combining genome-wide array and whole-exome sequencing (WES) data (*Belkadi et al., 2016*). In total, nine family members were genotyped with the Genome-Wide Human SNP Array 6.0. Genotype calling was achieved with the Affymetrix Power Tools Software Package (http://www.affymetrix.com/estore/partners_programs/programs/developer/tools/powertools.affx). SNPs were selected with population-based filters (*Purcell et al., 2007*), resulting in the use of 905,420 SNPs for linkage analysis. WES was performed as described in the corresponding section, in four family members, P1, P2, P3 and P4. In total, 64,348 WES variants were retained after application of the following filtering criteria: genotype quality (GQ) >40, minor read ratio (MRR) >0.3, individual depth (DP) >20 x, retaining only diallelic variants with an existing RS number and a call rate of 100%. Parametric multipoint linkage analysis was performed with the Merlin program (*Abecasis et al., 2002*), using the combined set of 960,267 variants. We assumed an AD mode of inheritance, with a frequency of the deleterious allele of 10$^{-5}$ and a penetrance varying with age (0.8 above the age of 65 years, and 0.02 below this threshold). Data for the family and for Europeans from the 1000 Genomes project were used to estimate allele frequencies and to define linkage clusters, with an r$^2$ threshold of 0.4.

The method used for WES has been described elsewhere (*Bogunovic et al., 2012*; *Byun et al., 2010*). Briefly, genomic DNA extracted from the patients' blood cells was sheared with a Covaris S2 Ultrasonicator (Covaris). An adapter-ligated library was prepared with the Paired-End Sample Prep kit V1 (Illumina). Exome capture was performed with the SureSelect Human All Exon kit (71 Mb version - Agilent Technologies). Paired-end sequencing was performed on an Illumina Genome Analyzer IIx (Illumina), generating 72- or 100-base reads. We used a BWA-MEM aligner (*Li and Durbin, 2009*) to align the sequences with the human genome reference sequence (hg19 build). Downstream processing was carried out with the Genome analysis toolkit (GATK) (*McKenna et al., 2010*) SAMtools (*Li et al., 2009*), and Picard Tools (http://picard.sourceforge.net). Substitution calls were made with a GATK UnifiedGenotyper, whereas indel calls were made with a SomaticIndelDetectorV2. All calls with a read coverage <2 x and a Phredscaled SNP quality <20 were filtered out. Single-nucleotide variants (SNV) were filtered on the basis of dbSNP135 (http://www.ncbi.nlm.nih.gov/SNP/) and 1000 Genomes (http:browser.1000genomes.org/index.html) data. All variants were annotated with ANNOVAR (*Wang et al., 2010*). All *IRF4* mutations identified by WES were confirmed by Sanger sequencing.

## Tw detection

PCR and serological tests for Tw were performed as previously described (*Fenollar et al., 2009*).

## Cell culture and subpopulation separation

PBMCs were isolated by Ficoll-Hypaque density centrifugation (GE Healthcare) from cytopheresis or whole-blood samples obtained from healthy volunteers and patients, respectively. PBMCs and EBV-B

cells (purified B cells were immortalized with EBV in the laboratory) were cultured in RPMI medium supplemented with 10% FBS, whereas HEK293T cells (ATCC; CRL-3216) were cultured in DMEM medium supplemented with 10% FBS. Subsets were separated by MACS, using magnetic beads conjugated with the appropriate antibody (Miltenyi Biotec) according to the manufacturer's protocol. All cells used in this study were tested for mycoplasma contamination and found to be negative.

## Site-directed mutagenesis and transient transfection

The full-length cDNA of *IRF4* and *PU.1* was inserted into the pcDNA 3.1D/V5-His-TOPO vector with the directional TOPO expression kit (Thermo Fisher Scientific). *BATF anf JUN* were obtained from Origene companie (#RC207104 and #RC209804, respectively). Constructs carrying mutant alleles were generated from this plasmid by mutagenesis with a site-directed mutagenesis kit (QuikChangeII XL; Agilent Technologies), according to the manufacturer's instructions. HEK293T cells were transiently transfected with the various constructs, using the Lipofectamine LTX kit (Thermo Fisher Scientific) in accordance with the manufacturer's instructions.

## Cell lysis and western blotting

Total protein extracts were prepared by mixing cells with lysis buffer (50 mM Tris-HCl pH 7.4, 150 mM NaCl, 0.5% Triton X-100, and 2 mM EDTA) supplemented with protease inhibitors (Complete, Roche) and phosphatase inhibitor cocktail (PhoStop, Roche), 0.1 mM dithiothreitol DTT (Life Technologies), 10 µg/ml pepstatin A (Sigma, #P4265), 10 µg/ml leupeptin (Sigma, #L2884), 10 µg/ml antipain dihydrochloride (Sigma, #A6191) and incubating for 40 min on ice. A two-step extraction was performed to separate the cytoplasmic and nuclear content of the cells; cells were first mixed with a membrane lysis buffer (10 mM Hepes pH 7.9, 10 mM KCl, 0.1 mM EDTA, 0.1 mM EGTA, 0.05 % NP40, 25 mM NaF supplemented with 1 mM PMSF, 1 mM DTT, 10 µg/ml leupeptin, 10 µg/ml aprotinin) and incubated for 30 min on ice. The lysate was centrifuged at 10,000 x *g*. The supernatant, corresponding to the cytoplasm-enriched fraction, was collected and the nuclear pellet was mixed with nuclear lysis buffer (20 mM Hepes pH 7.9, 0.4 M NaCl, 1,mM EDTA, 1, mM EGTA, 25% glycerol supplemented with 1 mM PMSF, 1 mM DTT, 10 µg/ml leupeptin, 10 µg/ml aprotinin). Equal amounts of protein, according to a Bradford protein assay (BioRad, Hercules, CA), were resolved by SDS-PAGE in a Criterion TGX 10% precast gel (Biorad) and transferred to a low-fluorescence PVDF membrane. Membranes were probed with unconjugated antibody: anti-IRF4 (Santa Cruz, M-17) antibody was used at a dilution of 1:1000 and antibodies against GAPDH (Santa Cruz, FL-335), topoisomerase I (Santa Cruz, C-21), and/or lamin A/C (Santa Cruz, H-110) were used as loading controls. The appropriate HRP-conjugated or infrared dye (IRDye)-conjugated secondary antibodies were incubated with the membrane for the detection of antibody binding by the ChemiDoc MP (Biorad) or Licor Odyssey CLx system (Li-Cor, Lincoln, NE), respectively.

## EMSA

Double-stranded unlabeled oligonucleotides (cold probes) were generated by annealing in TE buffer (pH 7.9) supplemented with 33.3 mM NaCl and 0.67 mM MgCl₂. The annealing conditions were 100°C for 5 min, followed by cooling overnight at room temperature. After centrifugation at 3000 x *g* at 4°C for 30 min, the pellet was suspended in water. We labeled 0.1 µg of cold probe in Klenow buffer supplemented with 9.99 mM dNTP without ATP, 10 U Klenow fragment (NEB) and 50 µCi d-ATP-$^{32}$P, at 37°C for 60 min. Labeled probes were purified on Illustra MicroSpin G-25 Columns (GE Healthcare Life Sciences), according to the manufacturer's protocol. We incubated 10 µg of nuclear protein lysate on ice for 30 min with a $^{32}$P-labeled (a-dATP) *ISRE* probe (5′–gat cGG GAA AGG GAA ACC GAA ACT GAA-3′) designed on the basis of the *ISG15* promoter or the κB probe (5′- gat cGC TCT TTA TTT TCC TTC ACT TTG GTT AC-3′) described by Brass et al. in 1999 (**Brass et al., 1999**). For supershift assays, nuclear protein lysates were incubated for 30 min on ice with 2 µg of anti-IRF4 (Santa Cruz, M-17) antibody or anti-goat Ig (Santa Cruz) antibody. Protein/oligonucleotide mixtures were then subjected to electrophoresis in 12.5% acrylamide/bis-acrylamide 37.5:1 gels in 0.5% TBE migration buffer for 80 min at 200 mA. Gels were dried on Whatman paper at 80°C for 30 min and placed in a phosphor-screen cassette for 5 days. Radioactivity levels were analyzed with the Fluorescent Image Analyzer FLA-3000 system (Fujifilm).

## Luciferase reporter assays

The (*ISRE*)$_3$ reporter plasmid (pGL4.10[luc2] backbone, Promega #E6651), which contains three repeats of the *ISRE* sequence separated by spacers, was kindly provided by Prof. Aviva Azriel (Department of Biotechnology and Food Engineering, Technion-Israel Institute of Technology). The *AICE* reporter plasmid (pGL4.10[luc2] backbone, Promega #E6651) contains part of the *IL23R* promoter (−254 to −216). HEK293T cells were transiently transfected with the (*ISRE*)$_3$ reporter plasmid (100 ng/well on a 96-well plate), the pRL-SV40 vector (Promega # E2231, 40 ng/well) and a *IRF4* WT or mutant pcDNA 3.1D/V5-His-TOPO plasmid (Invitrogen #K4900-01, 25 ng/well or the amount indicated, made up to 100 ng with empty plasmid), with the Lipofectamine LTX kit (Thermo Fisher Scientific), according to the manufacturer's instructions. For the *AICE* assay, we used the same protocol, but with the addition of *BATF* and *JUN* expression plasmids (25 ng/well each). Cells were used 24 hr after transfection for the *ISRE* assay and 48 hr after transfection for the *AICE* assay, with the Dual-Luciferase 1000 assay system kit (Promega #E1980), according to the manufacturer's protocol. Signal intensity was determined with a Victor X4 plate reader (Perkin Elmer). Experiments were performed in triplicate and reporter activity is expressed as fold-induction relative to cells transfected with the empty vector. Negative dominance was assessed by performing the same protocol with the following modifications: (*ISRE*)$_3$ reporter plasmid (100 ng/well for a 96-well plate), pRL-SV40 vector (40 ng/well), +*IRF4*+ WT and mutant plasmids were used to cotransfect cells, with a constant amount of WT plasmid (25 ng/well) but various amounts of mutant plasmid (25 ng/well alone made up to 100 ng with empty plasmid or with the indicated amount, made up to 100 ng with empty plasmid) amounting to a total of 240 ng/well. For the *AICE* assay, the same protocol was used, but with the addition of *BATF* and *JUN* expression plasmids (25 ng/well each).

## Microarrays

For the microarray analysis of PBMCs, cells from six *IRF4*-heterozygous individuals (three patients, P1-P3; and three asymptomatic relatives, HET1-HET3) and six *IRF4* WT-homozygous individuals (four healthy relatives, WT1-WT4; and two healthy unrelated controls, C1-C2) were dispensed into a 96-well plate at a density of 200,000 cells/well and were infected *in vitro* with live Tw at a multiplicity of infection (MOI) of 1, or with live BCG (*M. bovis*-BCG, Pasteur substrain) at a MOI of 20, or were left uninfected (mock). Two wells per condition were combined 24 hr post-infection for total RNA isolation with the ZR RNA Microprep kit (Zymo Research). For the microarray on EBV-B cells, we used 400,000 cells from three *IRF4*-heterozygous mutation carriers and three WT individuals from the kindred for total RNA isolation with the ZR RNA Microprep kit (Zymo Research). Microarray experiments on both PBMCs and EBV-B cells were performed with the Affymetrix Human Gene 2.0 ST Array. Raw expression data were normalized by the robust multi-array average expression (RMA) method implemented in the affy R package (*Gautier et al., 2004*; *Irizarry et al., 2003*). Normalized expression data were processed as previously described (*Alsina et al., 2014*) and briefly summarized here. First, fold-changes (FC) in expression between mock-infected and BCG-infected or Tw-infected conditions were calculated for each individual separately. For each set of conditions, transcripts were further filtered based on a minimal 1.5 FC in expression (up- or downregulation). In a final stage, transcripts satisfying the previous filters in at least four of the six homozygous WT individuals for each *in vitro* infection condition were retained for downstream analysis. We counted the differentially expressed (DE) transcripts affected by stimulation in samples from all subjects for each stimulus, and determined the mean counts for these DE transcripts in all homozygotes. The mean values obtained were then used to normalize the counts of DE transcripts, yielding an overall transcriptional responsiveness for each individual separately, and for each stimulus. This overall responsiveness of subjects to either Tw or BCG is shown as a heatmap, and individual subjects were grouped by unsupervised hierarchical clustering. Responsive transcripts were further analyzed with Ingenuity Pathway Analysis (IPA) Software, Version 28820210 (QIAGEN) (*Alsina et al., 2014*) for functional interpretation. In brief, the FC values for each individual and treatment were used as input data for the identification of canonical pathway enrichment (*z*-score cut-off set at 0.1). The activation *z*-score values calculated for the identified pathways were exported from IPA and used to calculate mean values and differences between WT homozygotes and heterozygotes, and for graphical representation, with Microsoft Excel and GraphPad Prism Version 7.0, respectively. The direction of the difference was not considered further. Negative mean difference values were converted into positive values before the ranking of the canonical pathways according to the difference between the genotypes. The microarray data used in this study

have been deposited in the NCBI Gene Expression Omnibus (GEO) database, under accession number GSE102862.

## IRF4 qPCR

Total RNA was prepared from the EBV-B cells of individuals heterozygous for *IRF4* mutations (two patients and two asymptomatic relatives), and healthy homozygous WT relatives (*n* = 4). We also included samples from unrelated individuals (seven healthy controls and 25 patients with Tw carriage; all *IRF4* coding exons for each individual were sequenced and shown to be WT). Moreover, the 25 WD patients included in this experiment were found to have an intact *IRF4* cDNA structure and normal levels of IRF4 protein production in EBV-B cells. RNA was prepared from 500,000 cells with the ZR RNA Microprep kit (Zymo Research), according to the manufacturer's instructions. A mixture of random octamers and oligo dT-16 was used, with the MuLV reverse transcriptase (High-Capacity RNA-to-cDNA kit, Thermo Fisher Scientific), to generate cDNA. Quantitative real-time PCR was performed with the TaqMan Universal PCR Master Mix (Thermo Fisher Scientific), the *IRF4*-specific primer (Hs01056533_m1, Thermo Fisher Scientific) and the endogenous human  -glucuronidase (*GUSB*) as a control (4326320E, Thermo Fisher Scientific). Data were analyzed by the ΔΔCt method, with normalization against *GUSB*.

## IRF4 TA-cloning

The full-length cDNA generated from the EBV-B cells of *IRF4*-heterozygous and WT homozygous individuals was used for the PCR amplification of exon 3 of *IRF4*. The products obtained were cloned with the TOPO TA cloning kit (pCR2.1-TOPO TA vector, Thermo Fisher Scientific), according to the manufacturer's instructions. They were then used to transform chemically competent bacteria, and 100 clones per individual were Sanger-sequenced with M13 primers (forward and reverse).

## CD4+ T-cell stimulation by activating anti-CD2/CD3/CD28 mAb-coated beads

Isolated CD4+ T cells from patients (P1 and P3) and from healthy unrelated controls (C1-C4) were either left unstimulated (NS) or stimulated with activating anti-CD2/CD3/CD28 mAb-coated beads (Miltenyi Biotec) for 24 hr in RPMI medium supplemented with 10% FBS (24-well plate).

## Differentiation and activation of in vitro monocyte-derived macrophages (MDMs)

After isolation, monocytes from P1 or two healthy unrelated controls were plated (24-well plate; 600,000 cells/well) in RPMI medium supplemented with 40% human serum (M1-like) or 10% FBS (M2-like). Differentiation cytokines (R and D Systems) were immediately added: 0.5 ng/ml rhGM-CSF (M1-like) or 20 ng/ml rhM-CSF (M2-like). Every 3 days, we replaced 30% of the medium with fresh complete medium supplemented with the appropriate cytokines. After 14 days of differentiation, cells were left unstimulated (NS) or were activated by incubation with 2.5 ng/ml IFN-  (M1-like) or 50 ng/ml rh-IL-4 (M2-like) for 48 hr.

## MDM surface marker expression

Differentiated/activated MDMs were detached by treatment with trypsin (1.6 µg/ml) in PBS. Cells were treated with Fc receptor blocking agent (Miltenyi Biotec) and Aqua Dead Cell Stain kit (Thermo Fisher Scientific) for 1 hr. They were then washed and stained for 1 hr at room temperature with appropriate antibodies (see *Figure 5—figure supplement 4*) and appropriate isotype controls (BD biosciences). Samples were analyzed on a Beckman Coulter Gallios flow cytometer.

## Percentage of memory B cells

PBMCs from healthy unrelated controls and patients (P1, P2 and P3) were stained with antibodies against CD20, CD10, and CD27, and IgM, IgD, IgG, or IgA. The percentages of transitional (CD20+ CD10+ CD27-), naïve (CD20+ CD10- CD27-) and memory (CD20+ CD10- CD27+) B cells were determined

by flow cytometry. We then assessed the IgM/IgD or IgG or IgA expression of the memory B cells, to determine the extent of Ig isotype switching in the memory compartment.

## Ex vivo naïve and effector/memory CD4+ T-cell stimulation

CD4+ T cells were isolated as previously described (*Ma et al., 2012*). Briefly, cells were labeled with anti-CD4, anti-CD45RA, and anti-CCR7 antibodies, and naive (defined as CD45RA+ CCR7+ CD4+) T cells or effector/memory T cells (defined as CD45RA- CCR7± CD4+) were isolated (>98% purity) with a FACS Aria cell sorter (BD Biosciences). Purified naive or effector/memory CD4+ T cells were cultured with T-cell activation and expansion beads (anti-CD2/CD3/CD28; Miltenyi Biotec) for 5 days. Culture supernatants were then used to assess the secretion of IL-2, IL-4, IL-5, IL-6, IL-9, IL-10, IL-13, IL-17A, IL-17F, IFN , and TNF in a cytometric bead array (BD biosciences), and the secretion of IL-22, by ELISA (Peprotech).

## In vitro differentiation of naïve CD4+ T cells

Naïve CD4+ T cells (CD45RA+ CCR7+) were isolated (>98% purity) from healthy unrelated controls or patients, with a FACS Aria sorter (BD Biosciences). They were cultured under polarizing conditions, as previously described (*Ma et al., 2016*). Briefly, cells were cultured with T-cell activation and expansion beads (anti-CD2/CD3/CD28; Miltenyi Biotec) alone or under Th1 (IL-12 [20 ng/ml; R and D Systems]) or Th17 (TGF , IL-1  [20 ng/ml; Peprotech], IL-6 [50 ng/ml; PeproTech], IL-21 [50 ng/ml; PeproTech], IL-23 [20 ng/ml; eBioscience], anti-IL-4 [5 µg/ml], and anti-IFN-  [5 µg/ml; eBioscience]) polarizing conditions. After 5 days, culture supernatants were used to assess the secretion of the cytokines indicated, by ELISA (IL-22), or with a cytometric bead array (all other cytokines).

## Acknowledgement

We thank the patients and their families for participating in the study. We thank Yelena Nemirovskaya, Tatiana Kochetkov, Lahouari Amar, Cécile Patissier, Céline Desvallées, Dominick Papandrea, Mark Woollett, and Amy Gall for technical and secretarial assistance and all members of the Laboratory of Human Genetics of Infectious Diseases for helpful discussions. We acknowledge the use of the biological resources of the Imagine Institute DNA biobank (BB-33–00065). AG was supported by ANR-IFNPHOX (ANR-13-ISV3-0001-01), ANR-GENMSMD (ANR-16-CE17-0005-01) and the Imagine Institute. The Laboratory of Human Genetics of Infectious Diseases is supported in part by grants from the St. Giles Foundation, The Rockefeller University, Institut National de la Santé et de la Recherche Médicale (INSERM), Paris Descartes University, and the European Research Council (ERC), the Integrative Biology of Emerging Infectious Diseases Laboratory of Excellence (ANR-10-LABX-62-IBEID) and the French National Research Agency (ANR) under the 'Investments for the future' program (grant number ANR-10-IAHU-01), ANR-IFNPHOX (ANR-13-ISV3-0001-01, to JB), ANR-GENMSMD (ANR-16-CE17-0005-01, to JB). Research in the Quintana-Murci laboratory was supported by the Pasteur Institute, the *Centre National de la Recherche Scientifique* (CNRS), the French Government's Investissement d'Avenir program, (ANR-10-LABX-62-IBEID), IEIHSEER (ANR-14-CE14-0008-02) and TBPATHGEN (ANR-14-CE14-0007-02), and the European Union's Seventh Framework Program (FP/2007–2013)/ERC Grant Agreement No. 281297. SGT and CSM are supported by grants and fellowship awarded by the National Health and Medical Research Council of Australia (1113904, 1042925) and the Office of Health and Medical Research of the New South Wales State Government. TN and LW are supported by Australian Postgraduate Research Awards from the University of NSW.

## Additional information

### Funding

| Funder | Grant reference number | Author |
| --- | --- | --- |
| National Institutes of Health | 5R01AI089970-02 | Jean-Laurent Casanova |

| Funder | Grant reference number | Author |
|---|---|---|
| National Health and Medical Research Council | 1042925 | Cindy S Ma<br>Stuart G Tangye |
| European Research Council | 281297 | Lluis Quintana-Murci |
| Seventh Framework Programme | FP/2007-2013 | Lluis Quintana-Murci |
| Agence Nationale de la Recherche | ANR-IFNPHOX (ANR-13-ISV3-001-01) | Jacinta Bustamante |
| Agence Nationale de la Recherche | ANR-10-LABX-62-IBEID | Lluis Quintana-Murci<br>Laurent Abel<br>Jean-Laurent Casanova |
| Agence Nationale de la Recherche | ANR-GENMSMD (ANR-16-CE17-0005-01) | Jacinta Bustamante |
| Agence Nationale de la Recherche | ANR-14-CE14-0008-02 | Lluis Quintana-Murci |
| Agence Nationale de la Recherche | ANR-14-CE14-0007-02 | Lluis Quintana-Murci |
| National Health and Medical Research Council | 1113904 | Stuart G Tangye<br>Cindy S Ma |
| Agence Nationale de la Recherche | NKIR-ANR-13-PDOC-0025-01 | Vivien Béziat |

The funders had no role in study design, data collection and interpretation, or the decision to submit the work for publication.

## Author contributions

Antoine Guérin, Conceptualization, Data curation, Formal analysis, Validation, Investigation, Visualization, Methodology, Writing – original draft, Project administration, Writing – review and editing; Gaspard Kerner, Formal analysis, Validation, Investigation, Writing – review and editing, Software; Nico Marr, Formal analysis, Writing – review and editing, Resources; Janet G Markle, Conceptualization, Supervision, Writing – review and editing; Florence Fenollar, Conceptualization, Writing – review and editing, Resources; Natalie Wong, Data curation, Formal analysis, Validation, Visualization, Methodology, Writing – review and editing; Sabri Boughorbel, Formal analysis, Writing – review and editing, Software; Danielle T Avery, Formal analysis, Validation, Investigation, Writing – review and editing, Software; Cindy S Ma, Conceptualization, Formal analysis, Funding acquisition, Validation, Investigation, Writing – review and editing; Salim Bougarn, Formal analysis, Writing – review and editing; Matthieu Bouaziz, Formal analysis, Writing – review and editing, Software; Vivien Béziat, Conceptualization, Formal analysis, Funding acquisition, Writing – review and editing; Erika Della Mina, Conceptualization, Writing – original draft, Writing – review and editing; Carmen Oleaga-Quintas, Formal analysis, Writing – review and editing; Tomi Lazarov, Validation, Investigation, Writing – review and editing, Resources; Lisa Worley, Validation, Investigation, Writing – original draft, Writing – review and editing; Tina Nguyen, Validation, Investigation, Resources; Etienne Patin, Formal analysis, Writing – review and editing, Resources; Caroline Deswarte, Investigation, Writing – review and editing; Rubén Martinez-Barricarte, Formal analysis, Writing – review and editing; Soraya Boucherit, Formal analysis, Writing – review and editing, Resources; Xavier Ayral, Investigation, Writing – review and editing, Resources; Sophie Edouard, Validation, Investigation, Writing – review and editing, Resources; Stéphanie Boisson-Dupuis, Conceptualization, Writing – review and editing, Resources; Vimel Rattina, Formal analysis, Writing – review and editing, Resources; Benedetta Bigio, Writing – review and editing, Resources; Guillaume Vogt, Conceptualization, Supervision, Validation, Project administration, Resources; Frédéric Geissmann, Conceptualization, Writing – review and editing, Resources; Lluis Quintana-Murci, Funding acquisition, Writing – review and editing, Resources; Damien Chaussabel, Conceptualization, Formal analysis, Writing – review and editing; Stuart G Tangye, Conceptualization, Funding acquisition, Writing – review and editing, Resources; Didier Raoult, Investigation, Writing – review and editing, Resources; Laurent Abel, Conceptualization, Supervision, Funding acquisition, Validation, Writing – original draft, Writing – review and

editing; Jacinta Bustamante, Conceptualization, Supervision, Funding acquisition, Validation, Writing – original draft, Project administration, Writing – review and editing, Resources; Jean-Laurent Casanova, Conceptualization, Supervision, Funding acquisition, Validation, Writing – original draft, Project administration, Writing – review and editing

## Author ORCIDs
Antoine Guérin (iD) http://orcid.org/0000-0001-6810-1607
Gaspard Kerner (iD) https://orcid.org/0000-0003-0146-9428
Erika Della Mina (iD) http://orcid.org/0000-0001-8733-7623
Jacinta Bustamante (iD) http://orcid.org/0000-0002-3439-2482
Jean-Laurent Casanova (iD) https://orcid.org/0000-0002-7782-4169

## Ethics
Informed consent was obtained from all family members, and the study was approved by the national ethics committee.

## Decision letter and Author response
Decision letter https://doi.org/10.7554/eLife.32340.035
Author response https://doi.org/10.7554/eLife.32340.036

# Additional files

## Supplementary files
- Transparent reporting form.
DOI: https://doi.org/10.7554/eLife.32340.030

## Major datasets
The following dataset was generated:

| Author(s) | Year | Dataset title | Dataset URL | Database, license and accessibility information |
|---|---|---|---|---|
| Guérin A, Marr N, Boughorbel S, Bougarn S, Chaussabel D, Abel L, Casanova J | 2017 | IRF4 haploinsufficiency in a family with Whipple's disease | https://www.ncbi.nlm.nih.gov/geo/query/acc.cgi?acc=GSE102862 | Publicly available at the NCBI Gene Expression Omnibus (accession no: GSE102862). |

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
