## [Decision Letter]

Thank you for submitting your article "*IRF4* haploinsufficiency in a family with Whipple's disease" for consideration by *eLife* . Your article has been favorably evaluated by Harry Dietz (Senior Editor) and three reviewers, one of whom, Jos WM van der Meer (Reviewer #1), is a member of our Board of Reviewing Editors. The following individual involved in review of your submission has agreed to reveal their identity: Don Vinh (Reviewer #3).

The reviewers have discussed the reviews with one another and the Reviewing Editor has drafted this decision to help you prepare a revised submission.

Summary:

This paper represents a solid piece of clinical investigation. The investigators describe a rare loss-of-function mutation in *IRF4* , leading to an autosomal dominant probably selective immunodeficiency in the defense against *Tropheryma Whipplei* .

The role of this variant/gene in disease is supported by the authors:

a) The authors show that *IRF4* has evolutionary constraint against variation in the human population;

b) The affected residue is highly conserved and *in silico* prediction programs predict pathogenicity;

c) The authors provide additional experimental evidence for pathogenicity by introducing the identified variant (and others observed in the human population) in an overexpression system;

d) Use an EMSA assay to give clues about the pathophysiological consequences of the variant; and finally show RNAseq data with and w/o *in vitro* stimulations.

Essential revisions:

1) The work is convincing that this mutation underlies the susceptibility in this family. That being said, we come to the major weakness of the paper, i.e., the lack of confirmation in either a second family or in (some) sporadic cases. The authors have investigated transcription in a series of unrelated patients with Whipple's disease (Figure 4). They do not provide much information on these patients. The least they could have done was sequencing of the entire *IRF4* gene in these patients. In addition, the authors should take the effort to use Matchmaker exchange to increase the chances of identifying other cases.

2) The findings they present in fact show that not only WD is associated with the mutation, but also (long-term?) carrier status. This point is rather prominent, but not discussed until pretty late in the Discussion. The possibility that the defect merely leads to (asymptomatic) infection of the GI tract should be discussed more clearly. It is possible that a second hit is necessary to lead to symptomatic infection.

Since *T. whipplei* primarily enters through the GI tract, the question is relevant which cells in the intestine express *IRF4* . This could be added to the Discussion.

3) The claim that *IRF4* haploinsufficiency is the mutational mechanism is not clear enough. Haploinsufficiency, per se, requires that the gene product (RNA and/or protein) is present in only 50%. The authors show even higher RNA levels for *IRF4* variants and no difference for protein levels, which would argue against HI. The term "functional haploinsufficiency" might be applied to indicate that the protein derived from one allele has no function, but this would need to be distinguished from a hypomorphic or dominant-negative allele, both of which are also defined by loss-of-function. It is questionable whether overexpression experiments are conclusive for LoF variants and at all allow insights into haploinsufficiency. What about *IRF4* deletions (and clear cut stop-gains or frameshift mutations), in controls and cases (with other diseases)?

---

## [Author Response]

Essential revisions:1) The work is convincing that this mutation underlies the susceptibility in this family. That being said, we come to the major weakness of the paper, i.e., the lack of confirmation in either a second family or in (some) sporadic cases.

It is true that we describe a single kindred with WD due to *IRF4* haploinsufficiency. However, this kindred is multiplex and large, and includes four distantly related patients. The first genetic etiology of primary immunodeficiencies has often been found in a single kindred, and even in a single patient (see Casanova JL et al., J. Exp. Med., 2014). The three previously reported inborn errors of human IRFs were each documented in a single patient: autosomal dominant *IRF3* deficiency causing herpes simplex encephalitis (Andersen LL et al., 2015), autosomal recessive *IRF7* deficiency causing life-threatening influenza (Ciancanelli M. et al., 2015), and autosomal recessive *IRF8* deficiency causing combined immunodeficiency (Hambleton S. et al., New Eng. J. Med., 2011). In this context, our multiplex kindred with WD and autosomal dominant *IRF4* deficiency provides a sufficient level of causality. It also provides an “anchor gene”, as other genetic etiologies of WD discovered in the future are likely to be related to *IRF4* . Indeed, a hallmark of the human genetic basis of infectious diseases to date is the high level of physiological homogeneity despite enormous genetic heterogeneity (see Casanova, 2015a and b). The manuscript has been revised accordingly (Discussion, first paragraph).

The authors have investigated transcription in a series of unrelated patients with Whipple's disease (Figure 4). They do not provide much information on these patients. The least they could have done was sequencing of the entire *IRF4* gene in these patients.

We thank the editors and referees for this helpful suggestion. We sequenced the coding and non-coding exons of *IRF4* , and the flanking intron regions, in 25 other unrelated patients with Whipple’s disease (WD). These findings are described in revised Figure 4. No rare or private *IRF4* variants were identified. We also searched for copy number variations (CNVs) encompassing parts of *IRF4* , but we found none. In addition, we amplified the full-length *IRF4* cDNA from EBV-B cell lines from the other 25 unrelated patients with WD. The structure of the cDNA was found to be intact in all these patients, ruling out regulatory mutations disrupting splicing of the*IRF4* mRNA (see Author response image 1).

**Author response image 1. respfig1:** Electrophoresis of the full-length IRF4 cDNA from EBV-B cell lines. EBV-B cell lines from 25 patients diagnosed with Whipple’s disease (WD1 to WD25, WT/WT at all coding exons of *IRF4*) and from three patients (P1, P2 and P3), two asymptomatic heterozygous relatives (HET1 and HET2), four homozygous WT relatives (WT1 to WT4) and two healthy controls (C1 and C2).

Finally, we analyzed IRF4 protein levels by western blotting on EBV-B cell lines from the other 25 unrelated patients with WD. We found that IRF4 levels were normal, strongly suggesting that these patients have no regulatory mutation decreasing the levels of *IRF4* transcript and protein (see Author response image 2). The manuscript has been revised accordingly (subsection “*IRF4* mRNA levels in transfected HEK293T cells”; subsection “*IRF4* qPCR”; Figure 4 legend).

**Author response image 2. respfig2:** IRF4 protein levels in EBV-B cell lines. Total lysates of EBV-B cells from five healthy controls (C1 to C5), four homozygous WT relatives (WT1 to WT4), two asymptomatic heterozygous relatives (HET1 and HET2), two patients (P1 and P3) and 25 other patients diagnosed with Whipple’s disease (WD1 to WD25, WT/WT at all coding exons of *IRF4* ). Protein extracts from HEK293T cells transfected with empty vector (E), WT or R98W plasmids were used as controls for the specific band corresponding to IRF4.

In addition, the authors should take the effort to use Matchmaker exchange to increase the chances of identifying other cases.

We thank the editors and reviewers for this suggestion. We used Matchmaker exchange and found no matches, for either the gene (*IRF4* ) or the clinical phenotype (WD). This was not completely unexpected, as WD is not even suspected to be a genetic disorder, and *IRF4* was not reported as a disease-causing gene.

2) The findings they present in fact show that not only WD is associated with the mutation, but also (long-term?) carrier status. This point is rather prominent, but not discussed until pretty late in the Discussion. The possibility that the defect merely leads to (asymptomatic) infection of the GI tract should be discussed more clearly. It is possible that a second hit is necessary to lead to symptomatic infection.

We thank the editors and reviewers for this important suggestion. We have revised our manuscript accordingly (Discussion, last paragraph).

Since T. whipplei primarily enters through the GI tract, the question is relevant which cells in the intestine express IRF4. This could be added to the Discussion.

We thank the editors and reviewers for this helpful suggestion. We have revised our manuscript accordingly (Discussion, last paragraph).

3) The claim that IRF4 haploinsufficiency is the mutational mechanism is not clear enough. Haploinsufficiency, per se, requires that the gene product (RNA and/or protein) is present in only 50%. The authors show even higher RNA levels for *IRF4* variants and no difference for protein levels, which would argue against HI. The term "functional haploinsufficiency" might be applied to indicate that the protein derived from one allele has no function, but this would need to be distinguished from a hypomorphic or dominant-negative allele, both of which are also defined by loss-of-function. It is questionable whether overexpression experiments are conclusive for LoF variants and at all allow insights into haploinsufficiency. What about *IRF4* deletions (and clear cut stop-gains or frameshift mutations), in controls and cases (with other diseases)?

We thank the editors and referees for raising this key question, which we have addressed experimentally in great depth.

We began by testing in more detail the hypothesis of negative dominance for the R98W allele. We first repeated the dominance negative (DN) assay using an (*ISRE* )_3_ promoter, testing a broader range of experimental conditions. We did not detect any negative dominance (revised Figure 3—figure supplement 1A), consistent with our original findings.

We then performed additional experiments using another type of promoter. IRF4 has been described to bind to promoters containing AP-1-IRF composite elements (*AICE* ) with a high affinity. IRF4 binds to the AP-1 complex in this way (Peng Li et al., Nature, 2012). We, therefore, assessed the ability of mutant IRF4 proteins to induce transcription from an *AICE* motif-containing promoter. As for the (*ISRE* )_3_ promoter, both R98W and R98A-C99A failed to activate the *AICE* motif-containing promoter (revised Figure 2D). R98W is, therefore, loss-of-function (LOF) for both (*ISRE* )_3_ and *AICE* motif-containing promoters. In this new context, we assessed the possible negative dominance of R98W using the *AICE* motif-containing promoter. We found no DN effect, even with high proportions of the mutant (revised Figure 3—figure supplement 1B). The R98W mutant allele is, therefore, LOF, as opposed to hypomorphic. Moreover, R98W is not DN, even when overexpressed, in any of the conditions tested. The R98W mutant allele being LOF but not DN, by definition, autosomal dominant IRF4 deficiency must operate by haploinsufficiency.

We then investigated the mechanism of haploinsufficiency in more detail. We found that IRF4 protein levels in the nuclei of EBV-B cells, where they would be expected to exert their effects on transcription, were similar in R98W heterozygous and WT homozygous individuals (revised Figure 4—figure supplement 1C). Similar results were obtained with nuclei from primary CD4^+^ T cells stimulated with activating anti-CD2/CD3/CD28 monoclonal antibody-coated beads (revised Figure 5C). Moreover, we have shown, by the TA cloning of amplified *IRF4* cDNAs from EBV-B cells, that cells from symptomatic and asymptomatic heterozygous individuals produce about 50% WT *IRF4* mRNA and 50% mutant *IRF4* mRNA (revised Figure 4B). Finally, we have shown that the overexpression of *IRF4* R98W or WT plasmids in HEK293T cells results in similar levels of IRF4 protein in the nucleus (revised Figure 3B). Together, these results indicate that IRF4 haploinsufficiency does not operate by loss-of-expression, but by LOF.

Finally, and importantly, we completed the functional characterization of all non-synonymous variants from the gnomAD database (revised Figure 3—figure supplement 4) or our in-house WES database (revised Figure 3—figure supplement 5). We had tested 39 of the 156 known variants by the time of our initial submission. For this revision, we recently tested a total of 153 variants. We did not test an essential splice variant (c.4032T>C) and two variants present only in a non-canonical transcript predicted to undergo nonsense-mediated decay (p.L406P and p.R407W).

The five stop-gain or frameshift variants predicted to be LOF in the gnomAD database were not detectable on western blots following the transfection of HEK293T cells with the corresponding plasmids (the epitope of the antibody used being located at the C-terminus of IRF4; revised Figure 3—figure supplement 2). We also performed RT-qPCR and amplified the full-length *IRF4* cDNA from HEK293T cells transfected with the corresponding plasmids. We found that the *IRF4* mRNAwas present and intact, confirming that the cells had been correctly transfected with the corresponding plasmids (see Author response image 3and Author response image 4).

**Author response image 3. respfig3:** IRF4 mRNA levels in transfected HEK293T cells. Total RNA extracted from HEK293T cells transfected with E, WT, R98W, R98A-C99A or the five predicted LOF variant plasmids, was subjected to RT-qPCR for total *IRF4* . Data are displayed as 2ΔΔCt after normalization against endogenous GUS control gene expression (ΔCt) and the WT value (ΔΔCt).

**Author response image 4. respfig4:** Electrophoresis of full-length IRF4 cDNA from transfected HEK293T cells. Total cDNA extracted from HEK293T cells transfected with E, WT, R98W, R98A-C99A or the five predicted LOF variant plasmids. A partial actin B cDNA was used as a loading control.

Overall, we showed that the five stop-gain or frameshift variants predicted to be LOF in the gnomAD database were LOF in functional tests in our overexpression system. All these variants are individually (MAF<6x10^-6^ ) and collectively (MAF <2x10^-5^ ) extremely rare, and we have no access to additional information for these individuals (a single heterozygote per variant) from the gnomAD database.

We also showed that, of the 148 non-synonymous coding variants tested, only one in-frame deletion (p.E46del) of one amino acid previously described in the gnomAD database was hypomorphic, and that only one in-frame deletion (p.G279-H280 del) of two amino acids found in our in-house WES database was LOF. The E46del is extremely rare in the general population (MAF= 9x10^-6^ ), and we have no access to additional information for the carrier of this *IRF4* variant in the gnomAD database. G279_H280 del has not been reported in public databases. The individual carrying this mutation does not suffer from WD and belongs to a family in which the allele does not segregate with the clinical phenotype.

Overall, the cumulative frequency of these seven LOF (*n* =6) or hypomorphic (*n* =1) variants (from a pool of 153 variants) is <4x10^-5^ , fully consistent with the frequency of WD (occurring only in adults chronically infected with Tw). The other 146 alleles behaved like the WT allele. These new findings confirm that IRF4 is intolerant to LOF, and even hypomorphic variants in the heterozygous state. They support our initial conclusion that autosomal dominant IRF4 deficiency underlies WD in this kindred through haploinsufficiency.

These new studies have resulted in several changes to the revised manuscript. The main text has been revised throughout. The following figures have also been modified:

- Revised Figure 3C: data are now shown as dose responses.

- Revised Figures 5A, 5B and 5C: data are no longer normalized relative to the mean value for controls.

- Revised Figure 2: as all variants have been tested, all dots are now in black.

- Revised Figure 3—figure supplement 2: we have added the new variants tested.

- Revised Figure 3—figure supplement 4: we have added the new variants tested.

- Revised Figure 4B: the data are now represented as a table.

Figures S3B and S3C, which showed the 39 variants previously tested have been removed, as 153 variants have now been tested, and the corresponding information is provided in revised Figure 2—source data 1.

Revised Figure 3D has been added. The previous Figures 3D and 3E are now revised Figures 3E and 3F, respectively.